# Exploring the topology and dynamic growth properties of co-invention networks and technology fields

**Pablo E. Pinto** *, **Guillermo Honores, Andrés Vallone**

Universidad Católica del Norte, Escuela de Ciencias Empresariales, Coquimbo, Chile

* ppinto@ucn.cl

**Data Availability Statement:** All relevant data are within the paper and its S1 Data files.

**Funding:** The author(s) received no specific funding for this work.

## Abstract

This study investigates the topology and dynamics of collaboration networks that exist between inventors and their patent co-authors for patents granted by the USPTO from 2007–2019 (2,241,201 patents and 1,879,037 inventors). We study changes in the configurations of different technology fields via the power-law, small-world, preferential attachment, shrinking diameter, densification law, and gelling point hypotheses. Similar to the existing literature, we obtain mixed results. Based on network statistics, we argue that the sudden rise of large networks in six technology sectors can be understood as a phase transition in which small, isolated networks form one giant component. In two other technology sectors, such a transition occurred much later and much less dramatically. The examination of inventor networks over time reveals the increased complexity of all technology sectors, regardless of the individual characteristics of the network. Therefore, we introduce ideas associated with the technological diversification of inventors to complement our analysis, and we find evidence that inventors tend to diversify into new fields that are less mature. This behavior appears to be correlated with the compliance of some of the expected network rules and has implications for the emerging patterns among the different collaboration networks under consideration here.

## Introduction

In a free market economy, patents have traditionally been considered an effective incentive scheme for the production of knowledge as they provide inventors the property rights to their inventions [1, 2]. By imposing legal exclusivity on the use of knowledge on a large scale, modern society encourages investment in research and development (R&D) therefore promoting innovation, economic growth, and greater social welfare [3]. However, the input (e.g., research expenditure) and output (e.g., patent applications) of innovation have significantly increased in technological fields over time and so the inventive-patent application participation has largely become a collective phenomenon, directly impacting the structure of inventor networks [4]. Therefore, the set of relationships between inventors is much more complex than has been previously thought. By associating with others, inventors do not form a simple

**Competing interests:** The authors have declared that no competing interests exist.

personal network but rather they become part of a wider (often global) and more complex network that appears to have produced a significant share of inventions worldwide. This trend can be attributed to communities [5], preferential attachment [6], small-world phenomena [7], and other factors (e.g., heavy tails for degree distributions), but it is also likely to be related to the characteristics of real large networks such as shrinking diameters and the densification power-law phenomena [8].

In what follows, we investigate the dynamic characteristics of eight technology field networks using patent data. We follow the existing literature [9] and compare the measurements of a series of static snapshots over time. On a large scale, patent networks seem to behave like other real networks, but they have certain characteristics that make them unique, and their level of maturity might be explained by differences in technology sectors. This paper therefore aims to contribute to an enhanced understanding of the dynamics governing the evolution of large technological networks using a social network analysis [10–12].

There have been tremendous advances in social network analysis (SNA) applied to large datasets over the last two decades [13]. SNA was developed based on graph theory and emerged when scientists noticed that the structure in which entities are connected to each other is sufficient to understanding specific behaviors [14]. According to SNA, the interactions between entities include actors (also called vertices or nodes) and ties (also called connections) [15]. Globalization has also led to a revolution in the availability of large datasets that capture major activities in knowledge fields and which can be analyzed through various SNA metrics [16]. A remarkable example of a large dataset is the patent data generated by the United States Patent and Trademark Office (USPTO), which includes the names of inventors (also referred to as the authors), the names of assignees, grant and application dates, home country, and detailed information regarding the technology classes and subclasses of over 10 million patents since 1790.

The present study investigates the topology and dynamics of collaboration networks between individual inventors and their patent co-authors for all patents granted by the USPTO from 2007 through 2019 (a total of 2,241,201 patents and 1,879,037 inventors). We construct a co-inventor network using the synchronized USPTO database and we analyze inventor collaboration events for the total network and by technology sector. The International Patent Classification (IPC) is used as it provides a hierarchical, well-structured system for the categorization of patents according to the different areas of technology to which they relate [17, 18]. The IPC divides patentable technology into eight major sections: A: Human necessities, B: Performing operations; Transporting, C: Chemistry; Metallurgy, D: Textiles; Paper, E: Fixed constructions, F: Mechanical engineering; Lighting; Heating; Weapons; Blasting, G: Physics, and H: Electricity. The assumptions underpinning our analysis are that each IPC category of patents represents the space of a specific technology field (as a network), and that the division between categories enables us to not only identify the characteristic structure of each technology field network [19], but also to map the inventor relationship data across different technology domains [20]. As suggested by [21], the organization of a patent map in relation to technology domains provides a missing link in the series of existing studies and will support the long-term goal of tracing innovations transversally through differently organized domains.

In this paper, we establish several hypotheses regarding the relationship between the structure and the dynamics of inventor networks, with a particular emphasis on technology domains. Following [9], the questions of interest under analysis include: What do co-invention networks across technology domains look like on a large scale? What sort of clustering behavior occurs? Do technology networks undergo a "phase transition", in which their behavior suddenly changes? What distributions and patterns do technology graphs maintain over time? To answer these questions, we study changes in the structural configurations of eight technology

fields (IPC) via the power-law, small-world, preferential attachment, shrinking diameter, densification law, and 'gelling point' hypotheses. Similar to the existing literature [22–26], we obtain mixed results. Based on network statistics, we argue that the sudden rise of giant networks in six sectors can be understood as a phase transition in which small, isolated networks form one giant component. Such a transition occurred much later and much less dramatically in two other sectors. The examination of inventor networks over time reveals the increased complexity of all technology sectors, regardless of the individual characteristics of the network. Therefore, we introduce ideas associated with the technological diversification of inventors [27, 28] to complement our analysis, and we find evidence that inventors tend to diversify into new fields that are less mature. This behavior appears to be correlated with the compliance of some of the expected network rules and has implications for the emerging patterns among the different collaboration networks under consideration here. Overall, we found that the relative growth rates of diversification paths for inventors patenting in all sectors have also stabilized after a period of time. This may be an indication that the inventor networks in all technology sectors are reaching a mature stage.

## Literature review and hypotheses

The role of inventor networks in the production of knowledge and innovation has long been recognized [29, 30], and has had a significant effect on productivity [31] and in the diffusion of knowledge [32]. Patents also leave behind a trail of interaction between inventors [33]. Recent work considers co-inventions (patents listing multiple inventors) as the practical expression of such collaborative research networks [34–36]. This has motivated the quantitative study of co-patenting networks, especially their statistical properties [37–39].

The network growth dynamics of technological fields have received less attention [27]. Historically, the core problem in studying innovations in different fields has been data availability, data silos and the limited access to cross-connections [40]. Today, the availability of large datasets that capture major activities in science and technology has created a revolution in the discipline of scientometrics [16]. One representative example is patent data produced by the USPTO, and particularly, the current organization of patent data in terms of IPC sections which provides the tools to address central questions of network analysis such as those derived in the study of different technological domains [21]. Many studies investigate technological and innovation activities by referring to links between nodes and the resulting networks [e.g., 17, 41–43]. However, the metrics identified (at both node and network level) are mostly discussed from an economic perspective in only one field of technology or one geographic region. This offers us an opportunity to study the variations in the emerging patterns among collaboration networks, specifically inventor networks, through the lens of differently organized technological domains [19–21, 44].

## Static properties

Models of complex networks can be split into two groups: static and dynamic [45]. Static models focus on a characteristic of the network measured at a single point of time and include a degree distribution analysis (i.e., the number of edges incident to each node), the distance between nodes (as measured by the shortest path length), and the community structure (i.e., the number of sub-graphs in a network) [46]. At the network level, the node degree distribution actually follows a power-law, i.e., the probability $P(k)$ of picking a node with exactly degree $k$ scales with $k^{-\gamma}$ where $\gamma$ is a constant $\gamma > 0$, and $P(k)$ is the fraction of nodes with degree $k$ [9]. This corresponds to a right-skewed (heavy-tailed) distribution. In such a right-skewed distribution few nodes dominate the system, and many have a low degree. In the context of co-

inventor collaborations, the power-law reflects the existence of a few inventors with a very high degree (patents, connections), and many others with a low degree [47]. Power-law like degree distributions have been reported in a wide range of collaboration networks (including co-inventions) with numerical values of the scaling exponents varying across countries and over time [37, 48, 49]. Evidence exists, however, showing slightly disturbed power law distributions [22], and also significant deviations from pure power-law distributions [11]. Less is known about the inventor population dynamics within and across technology domains, and their repercussions on technology development [20]. With that in mind, our first hypothesis is based on SNA metrics on the growth dynamics of inventor networks in different technological domains:

Hypothesis H1: the degree distribution for the co-inventor collaboration network, regardless of the technological domain, will follow a power-law, with $\gamma > 1$

The distance between nodes represents the number of steps or edges that must be traversed on the path from one node to another, and is often linked to the small-world phenomenon [7]. Broadly speaking, a network falls into the class of SW if it simultaneously exhibits high local clustering like regular networks and a shortest mean path length that is similar to a matched random graph [50]. The SW phenomenon shows that in real-world networks the degree is almost never normal or Poisson distributed, as expected in a random graph [14].

In general, inventor networks correspond to a family of networks that share certain aggregate properties regardless of many of their individual characteristics, including some that are typical of small-worlds [44]. Researchers have found mixed evidence regarding the real small-worldliness of co-inventorship networks, which may suggest that the SW effect is either less pronounced or non-existent in patent collaborations [24, 25, 39, 51]. This is due in part to the nature of the invention process, and the fact that search decisions for collaborator(s)/team members are not necessarily taken by the inventors themselves but rather the firm, i.e. the managers [4]. The literature suggests [44] that this may be due to the fragmentation of inventors´ networks and also the fact that said networks are small-worlds in some technology fields (e.g., Instruments, Industrial Processes, Chemicals & Materials and Pharmaceuticals & Biotechnology), while in other fields (e.g., Electrical Engineering & Electronics) they are not. Nevertheless, this study does not provide any conclusive evidence on whether inventor networks in other technologies have small-world structures, and therefore further research is needed. Therefore, our second hypothesis is:

Hypothesis H2: the co-inventor collaboration network, regardless of the technological domain, will exhibit properties typical of small-worlds, with small-coefficient $\sigma > 1$.

Finally, the community structure refers to distinct regions or sub-groups that are more densely connected internally than with the rest of the network [5]. Community detection algorithms have been used to detect these sub-graphs based on their links to others [13, 52]. The community detection problem is closely related to that of clustering and so the results from the application of such algorithms are comparable to those from more traditional quantitative measures such as modularity [9].

## Dynamic properties

Dynamic properties are typically studied by comparing the measurements of a series of static snapshots over time [9]. Earlier studies have mentioned the need to investigate the dynamics of evolving networks [11], and this continues to be important for advancing the literature [10]. Various mechanisms have been advanced including studies on preferential attachment [6], shrinking diameters and the densification power-law [8], and the 'gelling point' [9]. Here we briefly explain and study each of them within the framework of network theory.

Preferential attachment (PA) is the general principle of cumulative advantage or the rich-gets-richer mechanism, and states the rate at which a node acquires a new tie based on its degree (i.e., as a function of the number of ties it already has) [47]. PA has been used to explain observations in a wide variety of collaboration networks. Early measurements for the citation network [53] and science collaboration [54] claimed linear or close to linear preferential attachment ($\alpha = 1$). Later measurements using large datasets and patent information revealed more complex growth properties, from superlinear ($\alpha \sim 1.25$) to sublinear ($\alpha = 0.8 - 0.9$) PA [55]. A recent study, however, found a negative association between the net change of the number of inventors and the net change of the number of links (that is, not all of the new inventors collaborate with inventors that are already established in the network), a result that appears to challenge the PA idea [36]. PA results have implications for the global innovation system in terms of organization, growth, and hierarchy, and they also present challenges in regard to the generation of new technologies [10]. These PA results therefore deserve further study in different fields, leading us to our third hypothesis.

Hypothesis H3: the growth of the co-inventor collaboration network will be explained, regardless of the technological domain, by a sublinear PA regime, with $\alpha < 1$.

In a similar vein, [8] analyzed the temporal evolution of a wide variety of networks and measured the average degree and diameter over time. Upon making predictions about the dynamic change of network structures, they noticed that the average distance between nodes often shrinks over time, this in contrast to the conventional wisdom (based on the SW network model) that such distance parameters should increase gradually as a function of the number of nodes. This pattern can be attributed to three fundamentally related phenomena.

First, graphs exhibit an average distance between nodes that often shrinks as part of their evolutionary process until it reaches an equilibrium [8]. Several experiments have been performed to verify that the shrinkage of diameters is not intrinsic to datasets, including the effects of missing past data [46] and disconnected components [56]. They seem to confirm the shrinkage as an inherent property of networks, and therefore we explore this in our dataset:

Hypothesis H4: the diameter of the co-inventor collaboration network, regardless of the technological domain, will keep shrinking over time with the addition of new edges until it reaches an equilibrium.

Second, most real graphs evolve over time following the densification power-law or growth power-law, with the equation $|E|(t) \propto |N|(t)^{\beta}$, and where $|E|(t)$ and $|N|(t)$ denote the number of edges and nodes of the graph at time step $t$ with $\beta$ being the densification exponent [46]. A densification exponent value greater than 1 indicates that the number of edges grows super-linearly with the number of nodes as the graph densifies over time. [56] recently describe similar increasing trends on the average degree over time and the number of edges as a function of the number of vertices in several collaboration networks (including patents, citations, and affiliation networks) indicating that graphs become dense. This could suggest that the densification of graphs is an intrinsic phenomenon. The question remains whether this property applies equally to all technology domain networks, leading us to our fifth hypothesis:

Hypothesis H5: the co-inventor collaboration network, regardless of the technological domain, will densify over time with the addition of new nodes, with $\beta > 1$.

Third, many small, disconnected components in real networks will merge over time and within a few periods form a giant connected component (GCC). This relationship is referred to as the 'gelling point' [9]. At this point, the GCC keeps growing, absorbing the vast majority of the newcomer nodes, while the network diameter continues to steadily decrease beyond this point. This observation was found to be consistent with the emergence of invisible colleges within scientific networks containing more than 50% of the nodes [57], and more recently, with the formation of a knowledge flow network of mobile inventors among US firms,

institutions, and universities [58], the concentration of patents and citations around large corporations in Europe [18], and the dramatic aggregation in China of oligopolistic communities with key nodes taking central positions over time [39]. Additionally it seems to indicate a phase transition process in which small, isolated inventor networks end up forming one giant and connected component [24]. Therefore, the final hypothesis is:

Hypothesis H6: the co-inventor collaboration network, regardless of the technological domain, will exhibit a 'gel point', at which the diameter spikes and disconnected components gel into a giant component.

## Empirical setting, data and methods

### The dataset

We constructed the collaboration networks for every inventor who worked on patents granted from 2007 to 2019 using PatentsView R package. A significant advantage of using datasets from USPTO PatentsView is the accuracy of the disambiguated inventor and patent classifications [59]. For calculations, we use an unweighted matrix $W$, *where $w_{i,j} = 1$ if $w_{i,j} \neq 0$ $\forall i \neq j$, and $w_{i,j} = 0$* otherwise, the $i,j$th cell contains the number of patents that both inventor $i$ and inventor $j$ produced. This value is used as a collaboration intensity index [60]. By adding each row (or column) in the matrix, the number of collaborations that each inventor has generated (the degree) is calculated [37]. This also enables us to exclude non-collaborative inventors ($d = 0$) [30]. Finally, such information can be represented by a graph in which all pairs of co-inventors are connected by an edge labeled with the patent's ID [14]. This co-inventor network is considered a one-mode or unipartite projection of a bipartite or affiliation graph with different statistical properties from that of a unipartite graph, including clustering coefficients and path lengths [25].

The original dataset contains over 3.6 million patents ranging from 182,978 patents granted in 2007 to 392,618 in 2019. Following [19], we identify the characteristic structure of technology fields (IPC) and their collaborative networks in order to reveal the mechanisms that led to innovative development. Currently, the number of IPC sections ranges from one to eight (A–H), with 131 classes (3-digit level) and 646 subclasses (4-digit level). Following [21], we remove database mistakes (e.g., IPC sections I, J, and so on: 61 patents in total, and patents without section information: 333,664), and use IPC statistics for 8 sections, 100 classes, and slightly over 600 subclasses. Finally, we dropped patents that had only one author (1,090,757).

The resulting dataset contained patents awarded by the USPTO between 2007 and 2019. However, they were filed over a longer period (1969–2019). This creates an additional problem: drawing annual networks based on the 'granted' date instead of the 'filed' date is inaccurate, since patent applications filed in one year but granted in another will end up showing up in different networks. To solve this assignation problem, we use the filed date to build the networks for each year. For graphical and calculation purposes, we removed patents filed before 1999 as they were insignificant in number and percentage (1,238 patents or 0.06% of the total). Thus, our final dataset contains 2,241,201 patents and 1,879,037 inventors. Note that a patent may contain several technical objects and consequently be assigned to more than one section (in our case, 574,737 patents). These patents are counted in order to map each technological domain. In the final dataset, nearly half of (47.9%) inventors patented only once, but 96.2% of patents were made by repeat inventors with more than one patent. We could thus track diversification as repeat inventors patented in more than one domain over time.

### Network analysis

The centrality metrics that are used for the static properties of the graphs include degree centrality, betweenness centrality, clustering coefficient (CC), and $k$-core. The degree centrality of

a node is its number of neighbors and it can easily be computed in the graph by counting the number of links incident on a node. The average degree of the nodes reflects the compactness of the network: the higher the average degree, the denser the network [61]. The betweenness centrality measures how frequently a node lies along the pathway of another node and is the sum of the fraction of the shortest path through the node between any two nodes, considered over all pairs of nodes. As such, it measures the number of times a node acts as a bridge along the shortest path between two other nodes. The CC of a node is the ratio of the actual number of links between the neighbors of the node to that of the maximum possible number of links between the neighbors of the node. The k-core of a node is a maximal subnetwork in which each node has at least degree k [62].

The more "dynamic" properties were analyzed according to the following techniques. For the detection of community structure, we run the [5] algorithm and igraph package [63]. For the SW property [7], we compute data permutation tests for equivalent random networks [51] using the SW package of R [64]. For power-law degree distribution, we use the distribution of connections between inventors [10] and we implement the powerRlaw R package [65]. Finally, we use the R Package PAFit algorithm [66] to estimate the PA effect over time. The parameter testing was performed using R. The formulas are:

**H1:** The discrete mass function of a power law distribution is:

$$P(X = k) = \frac{k^{-\gamma}}{\xi(\gamma, k_{min})}$$

Where $\xi(\gamma, k_{min})$ is the is the generalized zeta function [67]. The maximum likelihood estimator for the $\gamma$ parameter is:

$$\hat{\gamma} \simeq 1 + n \left[ \sum_{i=1}^{n} ln \frac{k_i}{k_{min} - 0.5} \right]^{-1}$$

The estimation procedure and the algorithms are described in [65].

**H2:** The network small-worldness is quantified by the $\sigma$ coefficient, calculated by comparing clustering (C) and path length (L) of a given network to an equivalent random network with the same degree [50]:

$$\sigma = \frac{C/C_{random}}{L/L_{random}}$$

When $\sigma > 1$; $C > C_{random}$ and $L \approx L_{random}$, the network is small-world. To estimate the $\sigma$ we used the algorithms available in the "small.world" function of the "brainwaver" R package [68].

**H3**: Following [69] in the PA mechanism, the probability $P_i(t)$ that a node $v_i$ acquires a new edge at time t is proportional to a positive function, $A_{k_i}(t)$, of its current degree $k_i(t)$. The function $A_k$ is the attachment function which assumes a log-linear form of $k^\alpha$, with $\alpha$ is the attachment exponent. In the fitness mechanism the probability $P_i(t)$ that a node $v_i$ acquires a new edge depends only on the positive number $\eta_i$ that can be interpreted as the intrinsic attractiveness. In their combined form, the probability $P_i(t)$ is proportional to the product of $A_{k_i}(t)$ and $\eta_i$:

$$P_i(t) \propto k^\alpha \times \eta_i$$

We follow the estimated parameter $\alpha$, algorithms and procedures described in [69]

**H4** The diameter of a network is the longest of all the calculated shortest paths in a network. The diameter was estimated with the available algorithms in the "diameter" function of the igraph R package [63].

**H5** The densification power-law or growth power-law can be expressed as:

$$|E|(t) \propto |N|(t)^{\beta}$$

where $|E|(t)$ is the number of edges of a graph at time step t, $|N|(t)$ denote the number of nodes of the graph at time step $t$ and $\beta$ is the densification exponent [46]. Using ordinary least square, the $\beta$ parameter was estimated using the following model specification:

$$\log(|E|(t)) = \delta + \beta \, log(|N|(t)) + u$$

**H6** The 'gel point' was estimated as described in [9].

## Diversification analysis

To analyze diversification within the network, a bipartite network is built between inventors and sections. This network is materialized in an adjacency matrix $A_{n \times 8}$, which is indirectly filled using the patents as a link, in a way that if an inventor $i$ registers a patent in section A, the matrix $A_{i,A}$ assumes the value of 1. This matrix is valued with the number of patents per inventor per year (considering that an inventor can produce more than one patent in the same section/year). This also allows for multiple registrations of the same patent in different sections. This bipartite network is then transformed into a unimodal distribution of the form $S_{8 \times 8} = A'A$ so that each $s_{ij}$ reflects the number of patents that an inventor has generated in sections $i$ and $j$ every year. This new matrix $S$ enables us to calculate the proportion of diversified inventors within each section.

Although it is common in the literature to use IPC classes to measure the patent diversification levels of inventors either using a patent map [21, 27] or a concentration index such as the Herfindahl-Hirschman [28, 70], in this work we focus on those inventors who diversify their work between IPC sections (that is, inventors that register one or several patents in a one-year period into two or more different sections). This way of calculating diversification is more limited, since inventors who register patents in a single section are considered non-diversified even though they have registered a patent for multiple uses (i.e., second-level patent classes or subclasses within a particular patent class or first-level patent class). In that sense, our choice of non-diversification resembles that encountered in related technological diversification, while diversification is rather associated with unrelated technological diversification [71].

To generate an index of diversification, the matrix $S$ is normalized by the total sum of the rows. In this way, each element $s_{ij}$ shows the participation (or probability, given that they are frequencies) of an inventor registering a patent in $i$ and $j$ sections. We use the mobility index [72] as a proxy, which is calculated as follows: $Pr = 1 - s_{ii}$, where $s_{ii}$ is the $i$th element of the main diagonal of the matrix $S$. The $Pr$ index is a statistic that ranges between 0 to 1, where 0 corresponds to perfect specialization and 1 corresponds to perfect diversification.

To represent the observed frequencies of the diversification components of each sector we use a contingency table. In general, tables made up of $R$ rows and $C$ columns are considered. For $i = 1,..,R$ and $j = 1,..,C$, $p_{ij}$ represents the probability that a random observation belonging to a population under study will be classified in the $i$th row and $j$th column of the table. Denoted by $p_{i \bullet}$ is the marginal probability that an observation will be classified in the $i$th row of the table. Similarly, $p_{\bullet j}$ denotes the marginal probability that an observation will be classified in the $j$th column of the table. The sum of the probabilities of all the cells in the contingency table must add up to 1.

Finally, to evaluate how likely it is that any observed difference between the periods arose by chance, a Pearson's chi-squared test ($\chi^2$) is applied:

$$\chi^2 = \sum_i \sum_j \frac{S_{i,j}^{t+1} - S_{i,j}^t}{S_{i,j}^t}$$

Where $S_{i,j}^{t+1}$ are the values in the final period and $S_{i,j}^t$ the values in the initial period for a matrix $S$.

## Results

Table 1 reports the general properties of the network of inventors for patents granted by the USPTO for the total network, the largest component and for a few nested networks, each of them built by considering only the patents belonging to specific technological sections or fields, such as A. Human necessities, B. Performing operations and transporting, C. Chemistry and metallurgy, D. Textile and paper, E. Construction, F. Mechanical engineering, lighting, heating, weapons, and blasting, G. Physics, and H. Electricity.

Networks of inventors are highly fragmented and highly variable, with over 66,000 connected components in the Physics field but less than 5,200 in the Textile and paper field. The largest connected components between technological fields also differ significantly in size, with the largest connected component containing between 70% to 80% of the non-isolated inventors in the total and in some sections (C, G, and H), and between 10% and 25% in others (D and E). In between, we have sections A (62%), B (59%), and F (49%). Overall it seems that

**Table 1. Co-inventor network statistics by technological field (IPC) and total, 1999–2019.**

| | Section | | | | | | | | Total network | The largest component network |
|---|---|---|---|---|---|---|---|---|---|---|
| | **A** | **B** | **C** | **D** | **E** | **F** | **G** | **H** | | |
| Nodes (Vertices) | 423,755 | 500,171 | 399,057 | 30,013 | 79,202 | 249,352 | 854,234 | 675,443 | 1,879,037 | 1,484,760 |
| Edges (Links) | 1,360,356 | 1,261,412 | 1,462,078 | 65,374 | 162,787 | 581,696 | 2,722,328 | 2,243,650 | 6,742,143 | 6,264,427 |
| Degree centrality | 6.4 | 5.0 | 7.3 | 4.4 | 4.1 | 4.7 | 6.4 | 6.6 | 7.2 | 8.4 |
| Betweenness centrality | 0.007 | 0.010 | 0.013 | 0.006 | 0.003 | 0.015 | 0.019 | 0.011 | 0.010 | 0.010 |
| Clustering coefficient | 0.415 | 0.437 | 0.379 | 0.614 | 0.520 | 0.467 | 0.299 | 0.267 | 0.229 | 0.225 |
| $k$-core | 4.60 | 3.78 | 5.05 | 3.65 | 3.35 | 3.55 | 4.30 | 4.33 | 4.57 | 5.18 |
| Connected components | 46,523 (261,177; 147) | 58,251 (292,810; 195) | 27,330 (295,309; 294) | 5,135 (3,349; 1,965) | 15,442 (19,065; 2,034) | 34,433 (121,126; 233) | 66,305 (629,482; 126) | 46,254 (521,501; 53) | 124,609 (1,484,760; 100) | 1 (1,484,760) |
| Largest component | | | | | | | | | | |
| Modularity | 0.818 | 0.833 | 0.797 | 0.826 | 0.807 | 0.842 | 0.745 | 0.728 | 0.718 | 0.718 |
| Number of communities | 13,044 | 15,974 | 14,407 | 255 | 1,217 | 6,960 | 32,088 | 25,964 | 65,944 | 65,944 |
| Average community size | 20.02 | 18.33 | 20.50 | 13.13 | 15.66 | 17.40 | 19.62 | 20.08 | 22.51 | 22.51 |
| 2–4 inventor/ community | 9% | 8% | 7% | 9% | 12% | 9% | 10% | 10% | 10% | 10% |
| 5–9 | 24% | 24% | 24% | 34% | 28% | 27% | 23% | 24% | 22% | 22% |
| 10 or more | 67% | 68% | 69% | 57% | 60% | 64% | 67% | 66% | 68% | 68% |
| SW coefficient ($\sigma$) | 20.65 | 121.60 | 144.10 | 26.99 | 44.65 | 103.59 | 124.74 | 113.89 | 10,671 | 10,671 |

we are facing the emergence of a giant connected component (GCC) similar to that visualized by [7] in SW, but this feature appears to be less pronounced in smaller and apparently less evolved technological fields.

The difference in magnitude of these technology networks is emphasized even more by the community analysis. We use the edge betweenness community detection algorithm, a technique based on vertex betweenness centrality [5]. In the case of Physics and Electricity, the two largest fields, 32,088 and 25,964 groups or clusters are detected, each containing an average of about 20 inventors, whereas the fields of Construction and Textile and paper, the two smallest, have 1,217 and 255 communities and an average of 13 and 16 inventors, respectively. Despite this difference, the communities clearly reflect agglomeration around the intra-field networks, with approximately 20% of the largest groups containing more than 50% of the nodes. This also indicates the dominance of certain groups or clusters in all observed networks.

The overall GCC network has a strong modularity of $Q = 0.72$. Modularity has the same value as [58] in their US inventor mobility network study which is based on an analysis of patent documents, which indicates that most of the nodes keep in touch with only a limited number of organizations with whom they are willing or able to exchange inventors. The GCCs of the various technology fields also show strong modularity values at $Q = 0.79 \pm 0.05$.

We also use the number of $k$-cores, the largest subgraph where vertices have at least $k$ interconnections [73], to characterize the structure of co-inventor networks. According to $k$-cores, the technological fields with large numbers of $k$-cores are again, with the exception of section B, the networks that generate the most patents. This is in line with recent findings showing that networks with larger $k$-cores have more robust or denser social networks between co-inventors than networks with a smaller number of k-cores, and this is significantly and positively related to the pace of invention [74]. The $k$-core result also confirms that the overall GCC exhibits a dense interconnected structure.

Fig 1 visualizes the layout for the total and IPC technology networks using Cytoscape's algorithm [75]. The Cytoscape software is designed to visualize very large networks [76]. The central organizing principle of Cytoscape is a network graph, with inventors represented as nodes and interactions represented as links or edges between nodes. The network core refers to a central and densely connected set of network nodes (the largest connected component of the network), while the periphery of the network denotes the sparsely connected set of nodes, which are linked to the core. The non-connected or isolated nodes from the bulk of the network are shown to the side and bottom of the central figure.

As illustrated above, differences in size and complexity are noticeable. The total graph includes 1.8 million vertices (inventors) and 6.7 million edges (links) in total. The largest connected component of the inventor network (1.4 million vertices and 6.2 million edges) is shown as a cloud in the middle. Fig 1G is an almost perfect replica of the total, even though it contains only 45% of the total vertices and 40% of the edges. The simplest graph is that of the network category of Textile and paper and is shown in Fig 1D. It has 30,013 vertices (2%), and 65,374 edges (1%).

To test the SW property, we compute permutation tests for equivalent random (r) networks (the same number of vertices and path length) using the SW package of R [64]. A real network has an SW structure if its average clustering coefficient is significantly higher than a random network [51]. As recommended by [77], we made long runs (1,000 data permutations) and computed the results. For the overall GCC, $p = 0$, with CC = 0.225; PL = 7.39; $CC_r = 0.0001$; $PL_r = 35.11$. The small-coefficient $\sigma$ is greater than 1 ($\sigma = 10,671$; with SW = 0.030). As such, the GCC has characteristics of SW, thus giving support for the hypothesis H1. We also determine the probability of randomly finding a network with a higher clustering coefficient for all sections. Our results consistently support the SW model of Watts and Strogatz over time and for all the technology fields (see Table 1 above).

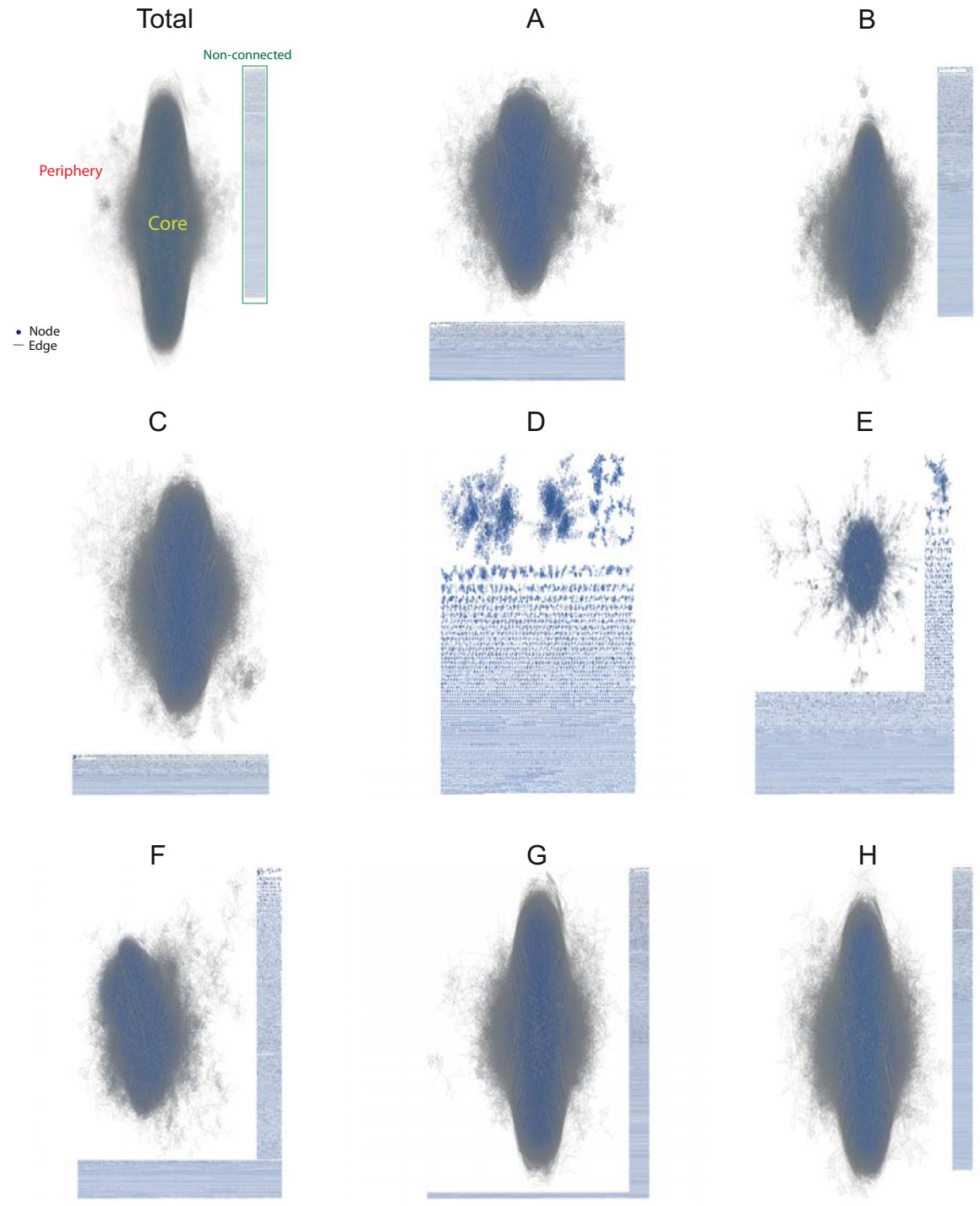

**Fig 1. Visualization of the technology co-inventor networks using Cytoscape, 1999–2019.**

Up until this point, we have examined the static properties of these networks. The literature suggests the use of temporal evolution into network analysis to significantly improve the quality of the results [78], therefore we examine several dynamic properties by comparing the measurements of a series of static snapshots over time [9].

We therefore turn to the second and third hypotheses, the power-law and PA properties. Fig 2 shows a scatter plot of the distribution of inventors according to the number of connections that they have. The data is plotted for 1999, 2009 and 2019 in order to investigate the inter-temporal stability of these networks [10].

Fig 2 illustrates the relative stability of the power-law exponents for the total network, particularly for 2009 and 2019. The power law (i.e., data can be fitted on a straight line on a log-log scale) behavior of the distribution of connections indicates that the total network follows a scale-free distribution [10]. However, the distribution appears to show what is called [11] "fat tails" (at the lower right-hand side) and "hooks" (at the top left-hand side), which may also suggest that the power-law operates only in the middle. This complicates the process of fitting the data to a straight line.

To check that the data came from a power-law model, we perform a goodness-of-fit test against a set of random synthetic power-law distributions based on the Kolmogorov-Smirnov (KS) statistics over 1,000 runs [48]. The scaling exponent ($\gamma$) indicates the presence of power-law for the total network; i.e., $\gamma = 3.49$ in 2019 ($p > 0.1$). In addition to the total network, some of the real graphs we studied also obeyed the power-law (namely: B. Performing operations and transporting, C. Chemistry and metallurgy, F. Mechanical engineering, lighting, heating, weapons, and blasting) with exponents ($\gamma$) between 4.64 and 4.91 for the year 2019 (data accumulated). Others (namely: A. Human necessities, E. Construction, D. Textile and paper, G. Physics, and H. Electricity) fail the test.

An interesting fact is that some of the graphs will exhibit power-law properties over the years, whereas others will not. For instance, the total network does not exhibit a scale-free range in the degree distribution in 1999. This can be explained because our empirical data contain only those nodes and links that have been created that year. By 2009 (as displayed in Fig 2), the distribution is already power-law ($\gamma = 4.37$) and will stay the same in subsequent years. Section C is the only network that exhibits power-law distributions in both 2009 and 2019. Section A shows power-law behavior in 2009 but not in 1999 or 2019. Compared to the other distributions (and with an average confidence level of 99%), sectors A y H behave like an exponential, sectors D and E behave like a Poisson, and sector G behaves like a lognormal model. Therefore, and based on our data, we found indications of scale-free networks, but also significant deviations from an ideal power-law in some of these networks over time. Thus, the hypothesis H2, which states that the co-inventor collaboration network will follow a power-law regardless of the technological domain, is not supported by our analysis.

As [14] points out, the concept of "scale-free" is particularly broad, and encompasses work that can either apply to a network with a power-law degree distribution or to a network that was specifically created by the preferential attachment model. Therefore, we also test whether PA is at work using the PAFit method [66]. Results can be seen from the plot of data points in Fig 3 with the horizontal axis showing the vertex degree $k$ value, and the vertical axis showing the attachment function estimate ($\alpha$). Only the case of $\alpha = 1$ gives rise to a perfect PA function [47]. For $0 < \alpha < 1$ the resulting degree distribution exhibits sublinear PA, whereas for $\alpha > 1$ superlinear attachment.

Our measurements using aggregated data from the category field networks show that the growth of co-inventorships can be explained based on the organizing principle of PA, although the attachment mechanism deviates from an ideal power-law. Applying SNA tools, our data reveals a sublinear preferential attachment model with exponent $\alpha = 0.67-0.72$. In the sublinear PA regime, new nodes (i.e., inventors) are connected to old ones with a probability proportional to a fractional power of their degree. In this case, the network becomes a gel-like where every node is connected to all other nodes and degree distribution is the stretched exponential rather than power-law [55]. This asymptotic degree distribution for sublinearly

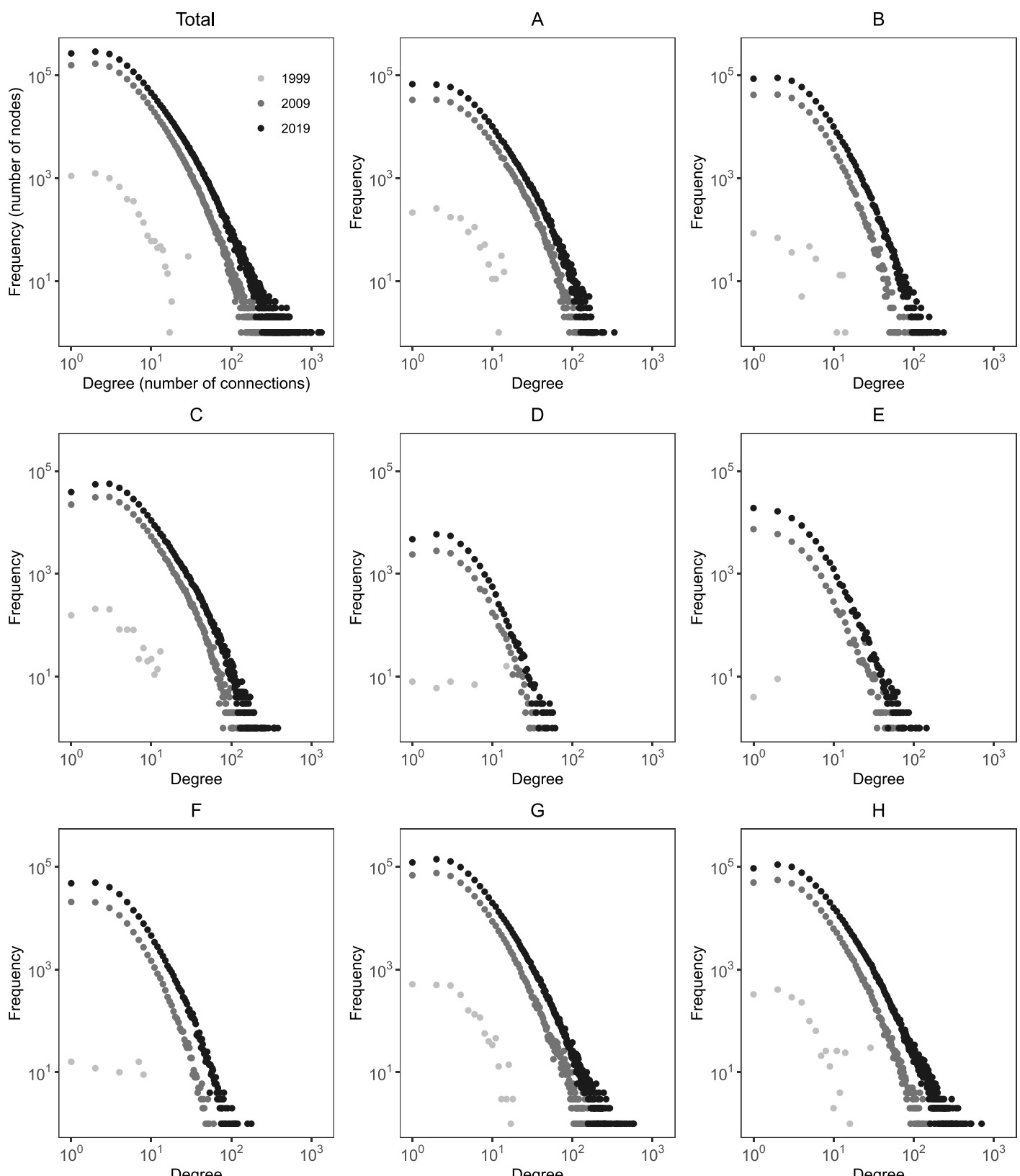

**Fig 2. Histogram: Distribution of inventors according to the number of connections by technological field, 1999, 2009 and 2019.**

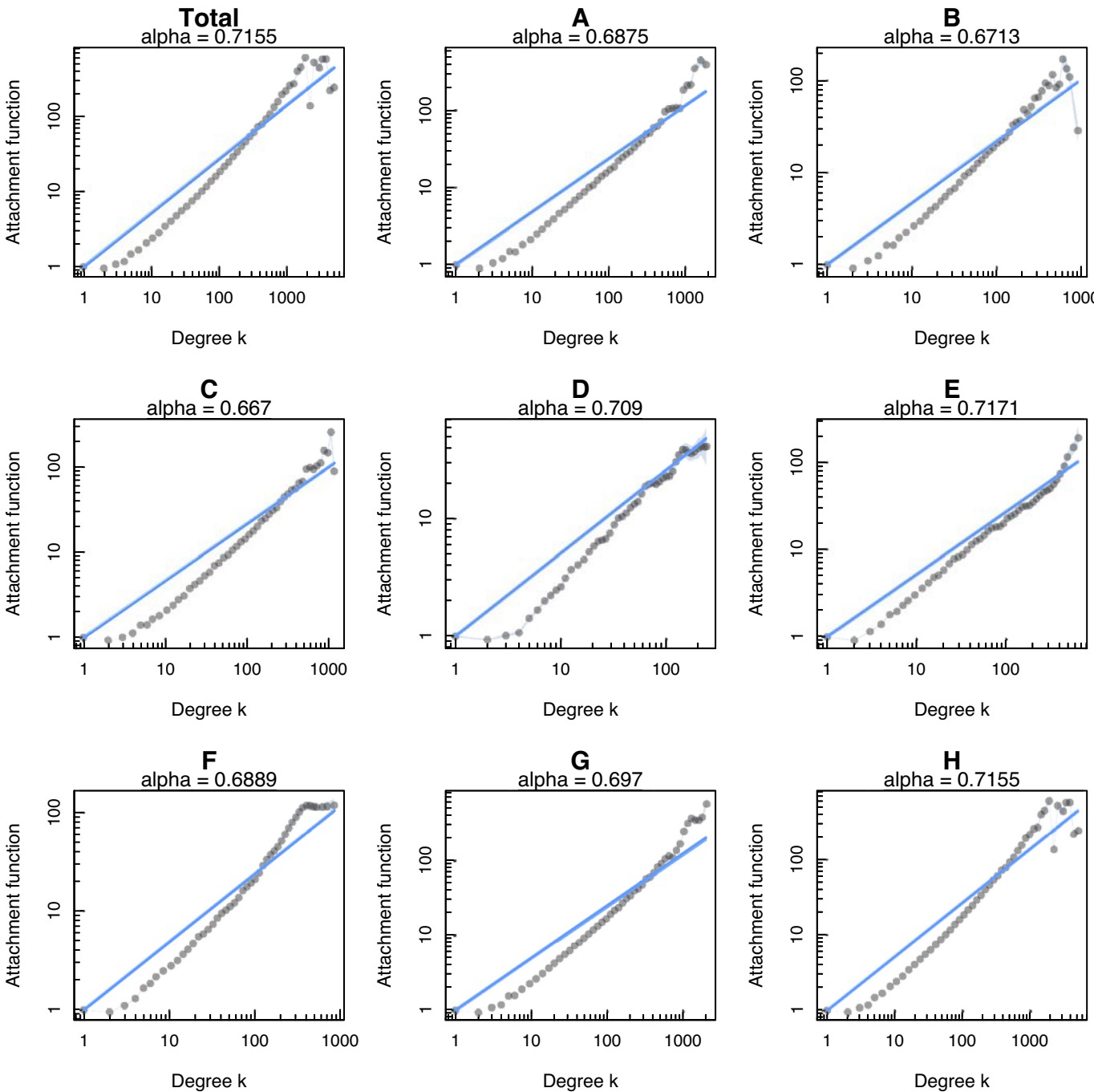

**Fig 3. The attachment function and node fitness by technological field using PAFit method.**

growing networks is characterized by an exponent smaller than one and a maximum degree that scales as a power of the logarithm of the number of nodes [79]. Therefore, our results from using large datasets support Hypothesis H3, that is, the sublinear regime, and this is consistent with previous measurements on scientific collaboration [11, 55]. This also indicates that a perfect PA function (that is, the chance that an inventor gets a new collaborator is proportional to their current number of collaborators) is not supported in our study.

We now turn to a review of the data regarding the final hypotheses. We first examine the shrinking diameter hypothesis via an analysis of the evolution of the total, each technology field network and the measurement of their diameters over time [8]. We next study the 'densification' and the 'gelling point' points in time at which the GCC absorbs the vast majority of the newcomer while the diameter of the network decreases or stabilizes [9].

The diameter, defined as the maximum distance between any two nodes, provides an indication of how much of a "small-world" the graph is (i.e., how quickly one can get from one "end" of the graph to another) [9]. In the context of our paper, this is an indication of how effective (or ineffective) technology field networks are in connecting pairs of inventors in the largest component. In line with [30], this can be attributed to each inventor in these fields working with more inventors than is normal in other fields. The diameter of the overall largest component is 31; therefore, 31 steps separate those inventors furthest apart from each other in this network. The path length, defined as the average number of steps along the shortest paths for all possible pairs of the network, is just 8.4. This indicates the SW property, with indirect links between inventors becoming remarkably short regardless of their location, industry or size of the network [58]. This is also consistent with our results showing overall decreasing values of diameter D and average path length L over time (D = 78 and L = 27.5 in 2002). Our results are in the range of those reported by other authors [30, 44].

The evidence we produced so far reveals that networks of inventors in all sections but E exhibit data consistent with decreasing diameters (Fig 4A). The results also indicate a large variation of diameter size between years and across IPC sections, however, networks have been shown to stabilize over time. By 2019, network diameters have reached between 30 to 33 in the most populated technological fields such as A. Human necessities, C. Chemistry and metallurgy, G. Physics, and H. Electricity, whereas in fields such as E. Construction, and F. Mechanical engineering, which are smaller in size, the prevailing diameter is still relatively large. These results are confirmed by looking at the number of links incident upon a node or degree centrality (reported in Table 1). The average degree centrality in four out of the five more populated technology fields is in the range of 6.4–7.3, compared to 4.1–4.7 in the least populated

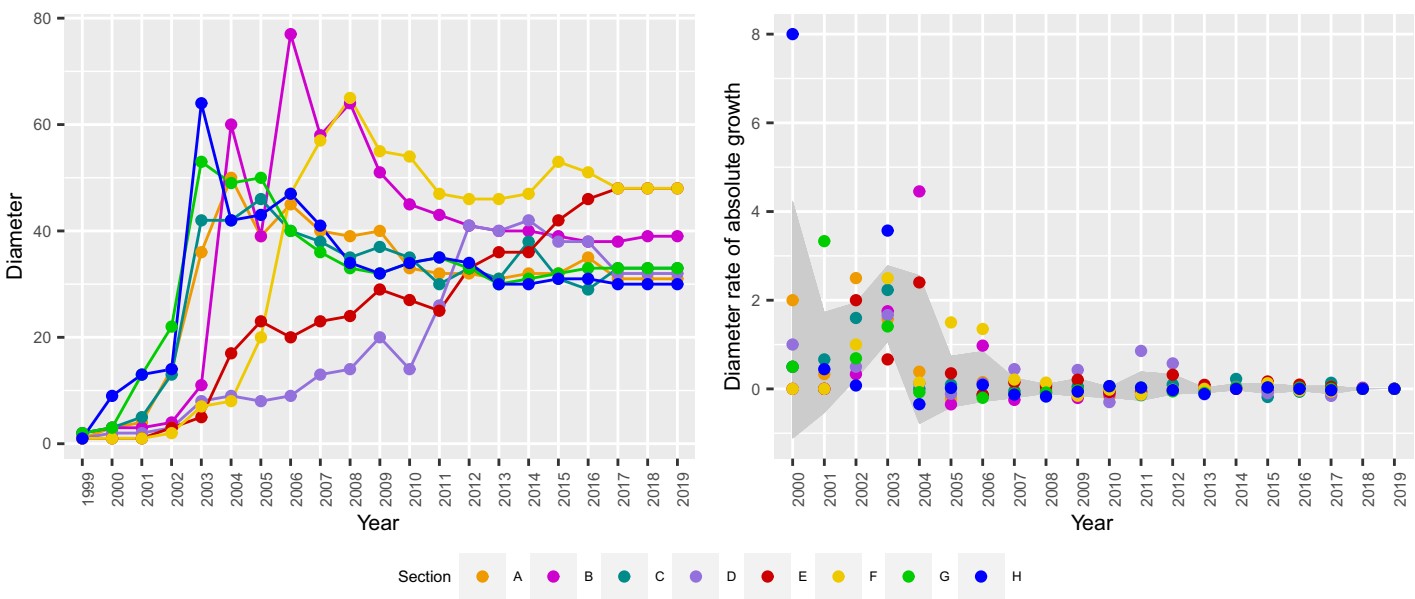

**Fig 4.** Diameter (left) and relative growth rate (right) over time by technological field, 1999–2019.

ones. The behavior of inventors in B. Performing operations/transporting, despite the much wider size of the network they are embedded in, is a relative exception, although the community of inventors there seems to have considerably tightened its relationships over time.

A graphic representation of the absolute growth rate over time (Fig 4B) shows size convergence among IPC sectors. The highlighted area in the graph represents the mean value of the growth rate per year plus/minus one standard deviation for all sectors over time. Both calculations, the zero-growth rate between technology sectors and the reduction of the variability rate over time, reinforce the idea of shrinking diameters followed by stabilization processes. Thus, based on our data, we conclude that the diameter of co-inventor collaboration networks shrinks over time in spite of the addition of new nodes, a phenomenon that can be observed in every technological domain although at different rates. Therefore, we accept hypothesis H4.

SNA studies have suggested that the shrinking diameter phenomenon can be attributed to the 'densification' [8] and the 'gelling point' [9] effects. To test the first effect and explore why we find differences among technology sections, we plot the number of nodes N(t) versus the number of edges E(t) in log-log scales over time and estimate the densification exponent $\beta$ (Fig 5). The total graph considered densifies over time with the number of edges growing linearly in the number of nodes, with $\beta = 0.97$ for the entire period from 1999 to 2019 and superlinearly with $\beta > 1$ from the year 2000 onwards ($\beta = 1.04-1.23$). The latter indicates that the number of edges E grows faster than the number of nodes N over time in this graph. This relation is referred to as the densification power-law (DPL) or growth power-law [46]. Similar results are found with other graphs. The relatively good linear fit ($R^2 \sim 1$) in a double logarithmic axis plot appears to also agree with the DPL, which indicates that the two variables are related through a power law of the type $|E|(t) \propto |N|(t)^{\beta}$. It also explains away the shrinking diameter phenomenon observed in our real graphs [9]. Therefore, we accept Hypothesis H5, which states that co-inventor collaboration networks will densify over time with the addition of new nodes, regardless of the technological domain.

Finally, to test the 'gelling point', we first study changes in the three largest clusters over time. The first interesting feature we notice is the sudden aggregation process in all but the D and E fields. Fig 6 shows the fraction of nodes that are part of the giant (GCC), the second (CC2), and third (CC3) largest connected component size over time. Following [46] we also set the post-year 2005 subgraphs with no past (i.e., for all nodes dated before $t_{2005}$, we delete all their links) in order to create the graph we would have gotten if we had only started collecting data at time $t_{2005}$. Because we delete the pre-$t_{2005}$ links, the size of the GCC is smaller, but, as the graph grows, the effect of these deleted links becomes negligible. In the full and post-$t_{2005}$ graphs we see the emergence of the giant component as the SW dominates evolution fairly quick, and they both converge to the same point. This indicates that the decrease is happening in a mature graph and not because many small disconnected components are being rapidly glued together [56].

We also notice that in most graphs, the GCC clearly dominates the rest, while CC2 and CC3 oscillate and then decrease to a low plateau. This coincides with the theory underpinning the gelling point phenomenon, which indicates that the GCC keeps growing, absorbing the vast majority of the newcomer nodes. Partial exceptions are sections D and E, with all three clusters showing some signs of growth. This suggests that the inventor networks in the Textile/paper, and Construction sectors do not yet exhibit a 'gelling point'. In part, this is because our data is limited in extension and also because both sectors are small in comparison to the rest, and therefore, it may be a sign of an immature, nascent innovation ecosystem. With this in mind, we are therefore able to present only partial evidence to support Hypothesis H6, which states that all co-inventor collaboration networks, regardless of the technological domain, will exhibit a 'gelling point', at which the diameter spikes and disconnected components gel into a

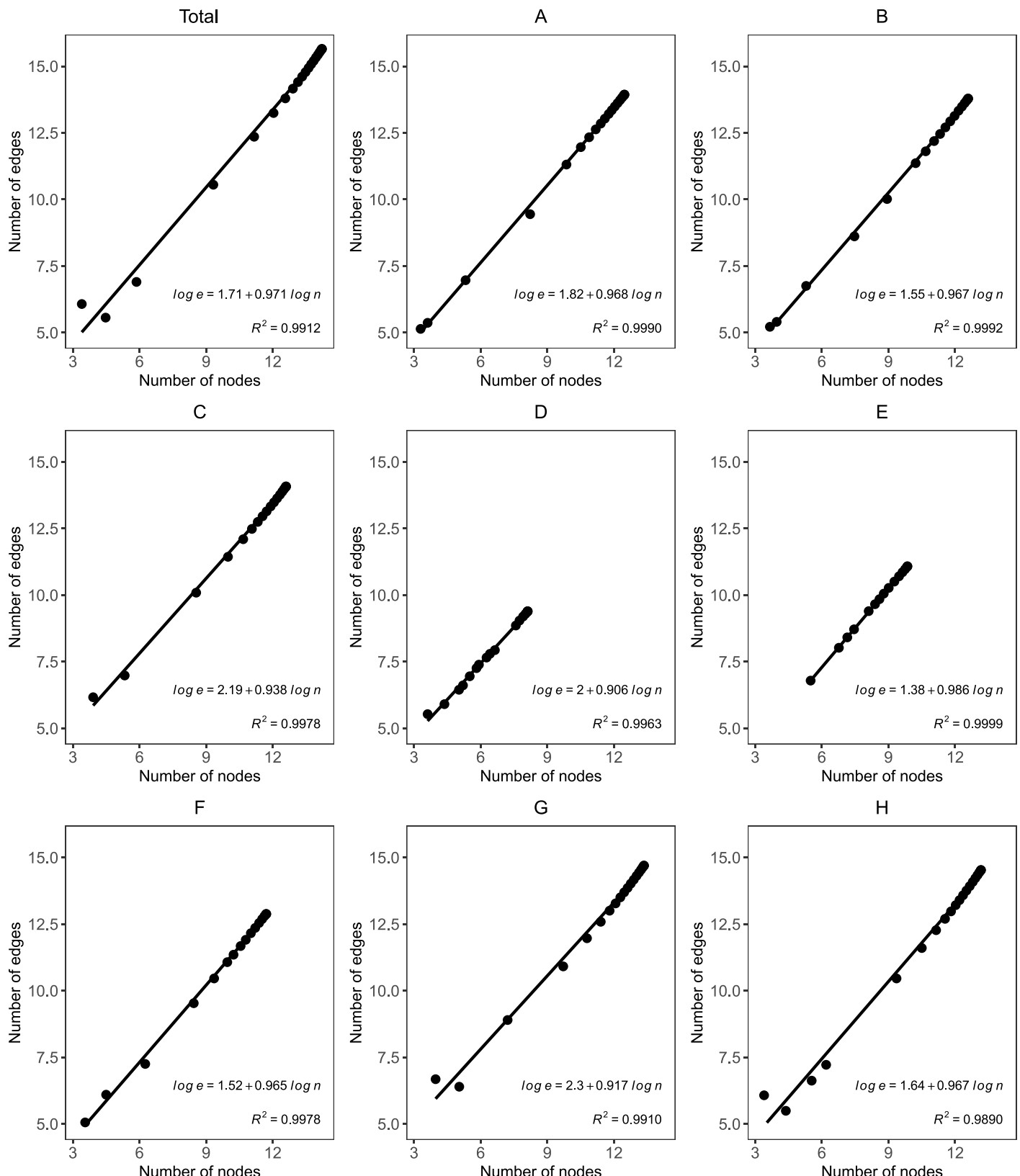

**Fig 5. The number of nodes N(t) versus the number of edges E(t) in log-log scales by technological field, 1999–2019.**

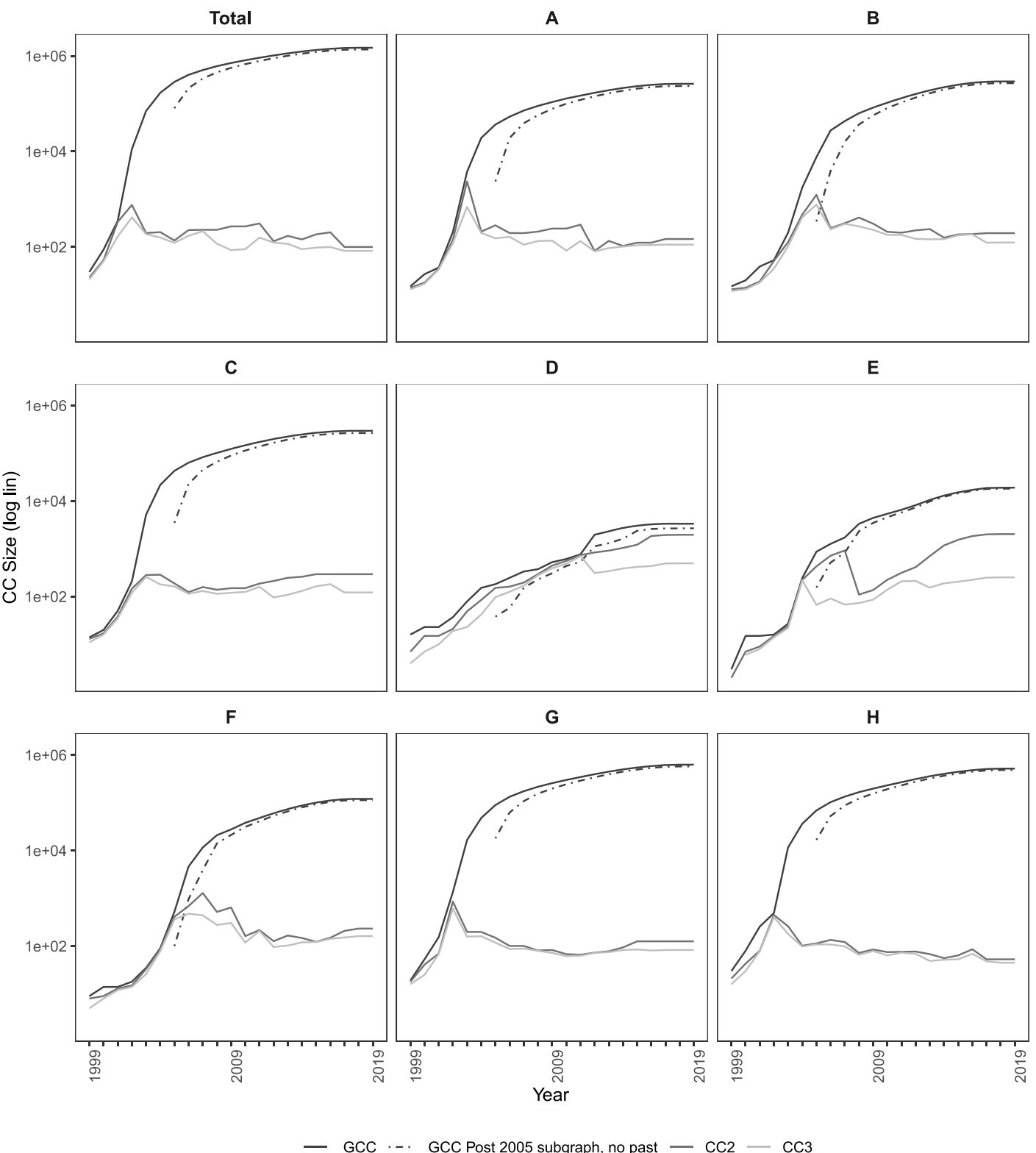

**Fig 6. The giant, 2nd and 3rd component size (in log) over time by technological field, 1999–2019.**

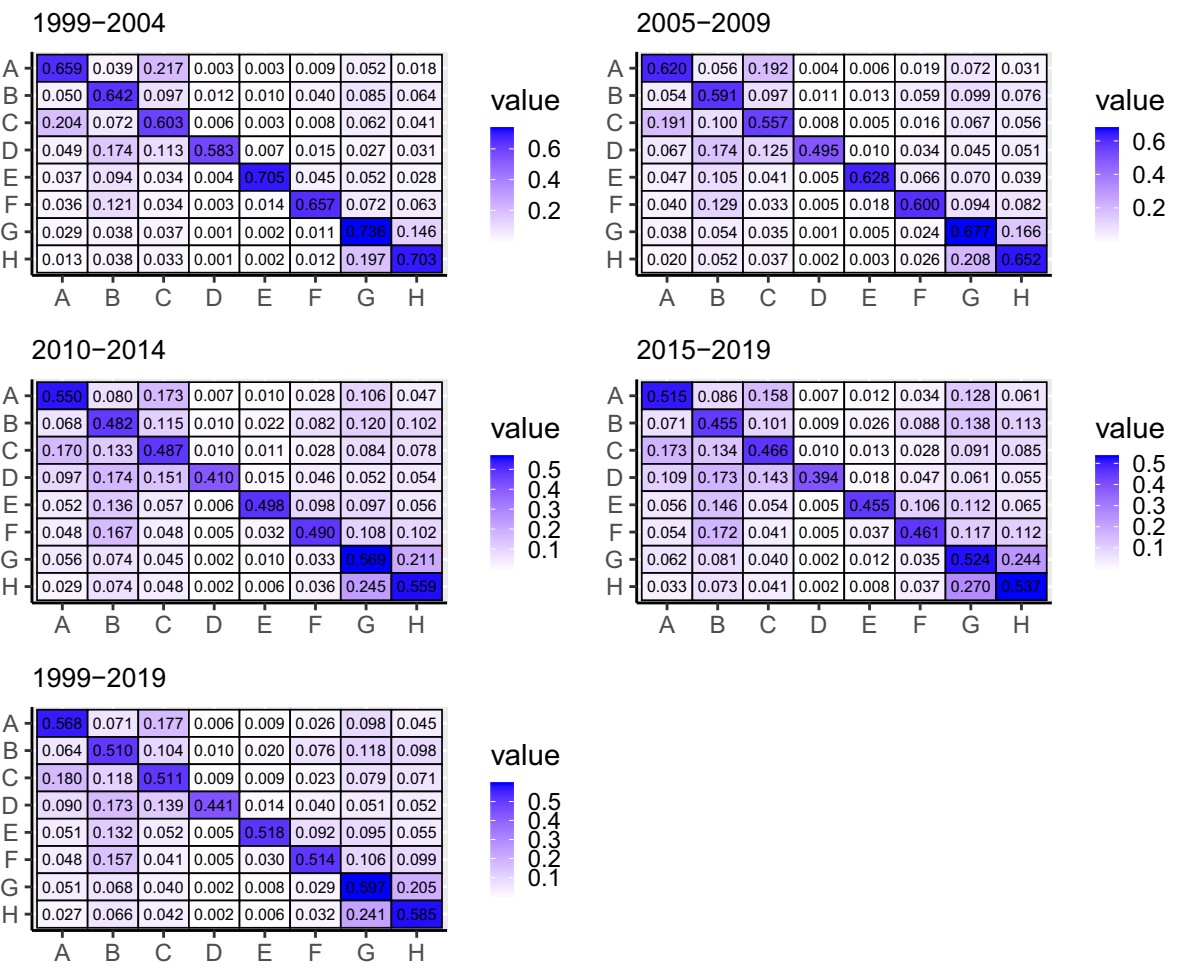

**Fig 7. Contingency tables over time by technological field, 1999–2019.**

giant component. However, we are certainly not able to refute the hypothesis based on our evidence.

Recent empirical studies have described the explanatory power of the diversification paths of inventors and inventor organizations on technology field network maps [27]. Therefore, one might speculate that the compliance/non-compliance with a network's rules in some of the real graphs observed here may be related to the inventor diversification pattern, among other factors. To analyze dynamic changes of each sector from 1999 to 2019, we also split data into four equivalent periods (data for the year 1999 was included in the first period) using a contingency table (Fig 7). For explanation purposes, we use the *Pr* index to rate diversification. Here, it corresponds to one minus the main diagonal value of the contingency table for each sector.

The static view (i.e., the average probability values over the total period) shows that the most specialized inventors correspond to those who have patented in A (*Pr* = 0.432), G (*Pr* = 0.403) and H (*Pr* = 0.415), with these two last sections also complementing each other relatively well (see highlighted areas). Sections A, B, C, D, E, and F have all average *Pr*-values between 0.482 and 0.490, whereas section D displays the highest diversification level, i.e., *Pr*>0.5. This indicates that at least 1 out of every 2 patents registered in section D is also registered in another one.

The more dynamic view indicates an increase in the diversification levels over time in all sectors, including in those sectors which were highly specialized at first (e.g., sectors E, G, and H). Inventors in sectors B, C, D, E, and F have also become more diversified than specialized surpassing the threshold $Pr>0.5$. They also appear to be patenting in various sectors at the same time.

While the analysis of the contingency tables shows a growing trend towards diversification, differences between periods can be considered subtle. Therefore, we tested how likely it is that any observed difference between the periods arose by chance using a Pearson's chi-squared test ($\chi^2$). The null hypothesis was rejected at a significance level of 1 per cent. Therefore, our study confirms that in the long run changes in diversification levels are statistically significant and they grow over time.

The contingency table also allows us to measure relationships between sectors. When analyzing cells other than the main diagonal ones, the $i, j$ values show the proportion of inventors who have jointly registered a patent. We can observe the set of relationships intensifies over time in some technological sectors. For instance, in the first observed period (1999–2004), 19.7% of inventors who registered patents in sector H also did so in G, and 14.6% of inventors who registered in G did so at the same time in H. In the last period (2015–2019), these numbers increased to 27.0%, and 24.4%, respectively. A similar relationship increasing over time is found with inventors patenting simultaneously in F and B, but with a lower intensity (a change from 12.1% to 17.2%). It is also possible to notice sectors with a stable relationship over time (e.g., between sectors D and B, with 17.4% of joint patents). Finally, there are also relationships that have lost intensity over time (e.g., A and C), although they remain relevant. Other sectors have few or almost no relationships (e.g., A, G, and H with D).

We observe the largest sectors (H, G, and B) showing an inflow greater than the smallest sectors (e.g., D and E). The data also allows us to detect a set of increasing relationships in the type, as is the case between G and H whose relationship increases period by period, and the case of section B, which is becoming more relevant to G and H. This reinforces the idea of both higher specialization levels and more focused diversification in the larger sectors.

## Discussion and conclusion

This study reveals distinctive dynamic growth features of inventor networks based on empirical evidence. We considered six hypotheses regarding the structure of global co-invention networks using data from the USPTO for patents granted between 2007 and 2019. Hypotheses were tested in a range of different networks and across several fields, and we found differential growth patterns spanning the different (IPC) technology sectors, with some following all the expected network rules, whereas others seem to be adjusting to those organizing principles. The findings of this study are summarized below.

First, we found evidence that technology field networks have a scale-free distribution. However, in some (namely: A. Human necessities, D. Textile and paper, E. Construction, G. Physics, and H. Electricity), the degree distribution deviated from an ideal power-law form. These networks appear to have fat-tailed degree distributions, but with some data points (at the start) clearly below the power-law straight line. This has been interpreted as a "power-law form" and (consistent with the literature) is characteristic of a young network, a small observation window and/or the influence of institutional constraints on the mechanism of self-organization [10, 11]. Taking into consideration the technology sectors involved, we can rule out the options of relatively new networks and a short window of time. Thus, technology networks, regardless of the sector, are distributed in a highly skewed manner rather than following a normal pattern.

Second, we also found that in spite of their fast expansion, all networks exhibited properties typical of small-worlds (all inventors in these networks are close to almost all other inventors through a small number of interconnecting ties). From an economic network perspective, more innovation is said to occur in highly clustered networks [60]. All the networks were highly clustered and fragmented into small communities of inventors, which is not unusual, and as [80] point out, a network can be both highly clustered and scale-free when small, and more extensive and less cohesive when it grows in size.

Third, all the inventor networks considered here grew at a lower PA rate than in other collaboration networks such as scientific papers and patent citations [55]. We can infer that the observed distributions fit in a growth model in which the source of added edges are chosen according to a sublinear PA, but the destination is selected at random [81]. This is indicative of an inverse Matthew effect or a cumulative disadvantage and implies that past patent activity is not necessarily correlated with whatever growth mechanism is actually at play [82]. Therefore, we cannot assume that there is a clear growth mechanism for making new connections for inventors with a high degree [81]. It is also possible that the innovation system is not yet at full capacity in terms of the number of players necessary to make it sustainable or self-organizing [54]. Self-organization may be among the most important mechanisms leading to scale invariance and decreased system entropy, which are the foundations of evolution in complex systems [82]. This sublinear PA regime seems to resonate deeply with the expected restrictions on the behavior of inventors in a business setting, in which inventors are required to first seek the collaboration of (or are coerced to collaborate with) others within their firms and corporate alliances before accessing new knowledge outside their organizational boundaries [4]. Patent collaboration is a complex task and company policies seem to play a decisive role, especially in shaping the partner selection of inventor teams and therefore collaboration networks. This is increasingly the case as the increasing regulatory patent protection system tends to restrict rather than advance knowledge spillover and sharing [26].

Patents provide evidence of the long-term capacity of a sector or industry to create novel technologies [15]. The traditional economic theory highlights the patent protection system, which by granting exclusive rights to inventors encourages *ex-ante* R&D investment, and through this contributes to increasing innovation and economic growth [2]. Under this traditional view, and as discussed by [3], patents as a policy instrument provide an elegant solution to the Arrow dilemma [1], which highlights the difficulty to achieve both optimal incentives to produce knowledge and an optimal level of dissemination of the produced knowledge. This theory, however, puts the emphasis on inventions as a solitary endeavor, and therefore ignores the collective nature of innovation. The literature indicates that many inventors work for research labs or research divisions of companies, and therefore in many instances an inventor's exploration of a domain is the result of a strategic managerial decision by the employer [20]. In a way, what we found here is that the patent system appears to restrict knowledge spillover, interaction, and collaboration (and therefore the sharing of knowledge and scientific advances) and thus call into question the traditional economic argument in favor of patent protection. This basic finding is in line with recent research and suggests that in a world in which innovations are largely assigned to private businesses, corporate and intellectual property restrictions appear to have a significant network effect on how inventors interact [26, 37]. The results of the present study, including our findings related to the degree distribution of inventor networks in young and mature industries, provide further evidence in support of this idea.

Fourth, our results also support the shrinking diameter and densification hypotheses for every technology sector, and we found partial evidence to support the gel point hypothesis, although some large and advanced fields' networks such as Physics and Electricity do exhibit data consistent with this theory, as does the total network. This is expected in more mature

networks since the number of nodes (inventors) remains constant over time, edges can only be added and not deleted, and densification naturally occurs [46]. This phenomenon is therefore consistent with critical transitions and the aggregation of isolated inventors as described in inventor network literature, and indicates that invention production, like any other knowledge production, undergoes a phase transition process during which small isolated inventor networks form one giant component [24].

The smallest technology groups, namely invention technologies in the fields of D. Textile and paper and E. Construction do not fit the gel point rules. Although there were indications of an early phase aggregation process, these inventor networks were still highly fragmented and much smaller in size than any of the other technology networks studied here, and this had important implications for their network topology. The creation of a complex structure as a result of the dynamics of the system is still pending in these sectors, as it is evident that the organization is still notably relegated in comparison with that of the other's sectors.

Finally, we explore a possible explanation by looking at the specialization/diversification patterns of inventors, and we found that inventors in all sectors over time have become more diversified. Overall, we found that the relative growth rates of diversification paths for inventors patenting in all sectors have also stabilized after a period of time. This may also be an indication that the inventor networks in all technology sectors are reaching a mature stage.

The reasons why inventor networks grow (investments, subsidies for collaborative research, government expenditures) or why inventors diversify (scientific advances, competition dynamics) go beyond pure "network laws". However, we find that in the long run that all types of technology networks appear to comply with these laws. There also appear to be signs of institutional restrictions on collaboration and patenting which appear to have reduced the size of the effects of some of these rules, especially in smaller and newer fields in the early periods of network formation. In summary, we show here that technology networks are complex and in the long run scale-invariant. This has important implications for the development of innovations over the different stages of technology networks.

This study has several limitations, all of which should be acknowledged for the sake of future research. First, the purpose of this study was to investigate the topology and the dynamics of collaboration between individual inventors in technology field networks. However, there are important factors which have been insufficiently discussed. For example, the depth of the co-inventor relationships or the influence of specific business settings on their collaborative behavior. Future studies need to triangulate patent information with other sources such as publications, patent citations, and/or firm data in order to explore other distinctive features of technology sector networks [for a recent study, see 18]. Future work can also consider a two-mode or bipartite inventor-assignee network where inventors are affiliated with organizations that are assigned the patent rights of an inventor in order to capture the influence of the organizational setting on the collaborative behavior of inventors. Another limitation is that our information was restricted to patents granted by the U.S. Patent and Trademark Office. This has a significant negative effect on the real size of the inventor networks in all technology fields, for example. Future research could benefit from including information on patents granted by other international patent offices to eliminate this bias [19]. The comparison with other studies is also limited by the fact that we use patent grants (rather than applications), which could have caused an overestimation of team size as single inventors are less likely to obtain a patent, or an underestimation an because not all inventors are listed on a patent application [37]. There are also large differences between technology classes with regard to the time period between applying and obtaining a granted patent as described in the text. Future research could take into consideration lag periods to account for a comparison between nascent and mature technology domains. Further limitations include the use of a much more

limited model of diversification than in previous studies [21, 27, 28, 71], and an insufficient discussion regarding inventor network evolution and diversification analysis. Future studies could also use econometric models and georeferentiation to examine technology crossover information over time and across territories.

## Supporting information

**S1 Data.**
(ZIP)

## Author Contributions

**Conceptualization:** Pablo E. Pinto, Andrés Vallone.

**Data curation:** Guillermo Honores, Andrés Vallone.

**Formal analysis:** Guillermo Honores.

**Investigation:** Pablo E. Pinto.

**Methodology:** Pablo E. Pinto, Guillermo Honores, Andrés Vallone.

**Project administration:** Pablo E. Pinto.

**Software:** Guillermo Honores, Andrés Vallone.

**Visualization:** Guillermo Honores.

**Writing – original draft:** Pablo E. Pinto, Andrés Vallone.

**Writing – review & editing:** Pablo E. Pinto.

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
