## [Decision Letter · Decision Letter 0]

24 Feb 2021

PONE-D-20-41087

Exploring the topology and dynamic growth properties of co-invention networks and technology fields

PLOS ONE

Dear Dr. PINTO,

Thank you for submitting your manuscript to PLOS ONE. After careful consideration, we feel that it has merit but does not fully meet PLOS ONE’s publication criteria as it currently stands. Therefore, we invite you to submit a revised version of the manuscript that addresses the points raised during the review process.

We look forward to receiving your revised manuscript.

Kind regards,

Petter Holme, Ph.D.

Academic Editor

PLOS ONE

Journal Requirements:

2. Please include captions for your Supporting Information files at the end of your manuscript, and update any in-text citations to match accordingly. Please see our Supporting Information guidelines for more information: http://journals.plos.org/plosone/s/supporting-information

Reviewers' comments:

Reviewer's Responses to Questions

**Comments to the Author**

1. Is the manuscript technically sound, and do the data support the conclusions?

Reviewer #1: Partly

Reviewer #2: Yes

Reviewer #3: Partly

Reviewer #4: Partly

2. Has the statistical analysis been performed appropriately and rigorously? 

Reviewer #1: Yes

Reviewer #2: Yes

Reviewer #3: Yes

Reviewer #4: Yes

3. Have the authors made all data underlying the findings in their manuscript fully available?

Reviewer #1: No

Reviewer #2: Yes

Reviewer #3: Yes

Reviewer #4: Yes

4. Is the manuscript presented in an intelligible fashion and written in standard English?

Reviewer #1: Yes

Reviewer #2: Yes

Reviewer #3: Yes

Reviewer #4: Yes

5. Review Comments to the Author

Reviewer #1: 1. This study only states the importance of the topic in the Introduction section and fails to identify whether relevant studies have been proposed previously, determine the difference between the present study and previous studies, or explain the innovativeness of the present study.

2. In the literature review section, this paper only explains what is known about the static properties and dynamic properties. This study did not review those related to the influence of different statistical properties on the productivity of the invention. The study was needed to spell out the contribution.

3. The development of the hypothesis requires sufficient theoretical or research support.

4. The figure is unclear and vague, such as Figure 8.

5. You can also put all the figures and tables at the end of the paper to avoid breaking up the text.

6. The discussion and conclusion of this study mostly explained the aforementioned the research hypothesis and did not draw relevant conclusions and management implications.

Overall, this study did not explain the influence of different statistical properties on the productivity of the invention that makes the research contribution of this article quite weak. The development of the hypothesis did not have sufficient theoretical or research support, nor did it explain its management implications.

Reviewer #2: The study of P.E. Pinto, G. Honores, and A. Vallones deals with statisticaland dynamic properties of the collaboration network associated with US patents. The authors perform a thorough and encompassing characterization of the network properties such as degree distribution,betweenness centrality, network diameter, growth dynamics, etc. for several technological fields and analyze them. Moreover, they show what these network properties tell us about the inner working of the these technological fields and how innovations appear there. This aspect of their work shall be commended since network analysis per se is an established topic now, and the challenge now is in its application to the analysis of real, nonmathematical problems.

The paper is written clearly, pedagogilcally, and I like it. However, regarding presentation- Figs. 2,3,5,8 are technically unsatisfactory. I can't read the values on X- and Y-axes, the text and data resolution are insufficient.

With respect to data analysis: Fig.3 shows plots that serve to demonstrate the preferential attachment. I do not understand how the authors plotted the data - the figure caption is insufficient. What is the time window for these dynamic measurements? With respect to analysisi, the data clearly can't be represented as dn/dt~n^(1+\\alpha) as the authors claim. The data for all fields seems to follow the linear dependence with offset, dn/dt~(n+n_0). Anyway, what is the meaning of this analysis with respect to the invention process? What does the linear or superlinear or sublinear trend tells us about onventor collaboration?

"gelling point" (??) or "gel point"?

What do the studies of network diameter tell us- are there many more new inventors or just the circle of collaborators increases?

Shrinking or non-shrinking of the network diameter- what are the implications for the invention process? Or, amy be these trends are the fingerprint of the new/old field?

I like the studies of diversification, but what do they tell us about dynamics of the patents in the field? Is new field less diverse than an old field? What does it tell us - the technological sector has reached a stationary state? Is it related to technological revolution? Does it tell us that the field recently passed through such revolution ao does it tell us that the field stagnates?

In summary, the authors perform a throrough network analysis and draw sporadic conclusions about the maturity, organizational restraints, and mode of cooperation in different technological fields. To my mind, it would be nice to supplement this by the analysis of the invention process in each field as follows from network analysis. This does not require new measurements- just a section in discussion where the already existing material is rearranged differently - not according to network properties, but rather to each technological field.

Reviewer #3: I would like to thank the authors for this very interesting paper and the rigorous way in which they conducted their analyses. I do have some issues with the paper, the main one being with the data used (see below). Besides that, I also make some recommendations to improve the paper and make it more convincing.

The paper is quite descriptive, describing the evolution of inventor networks over a given period. There is no clear research goal beyond describing the evolution of the network and as such it is unclear to the reader where the discussion about 'diversifying inventors' comes from and where it should be positioned in the literature. Is this an observation you stumbled upon or a discussion you want to engage in? In the latter case, you would have to include some theorization/hypotheses on this.

Furthermore, the authors mix theoretical concepts and measures/variables in the text (this starts at the introduction), I would strongly recommend to stick to 'concepts' in the first part of the paper and then move to the measures in the empirical part of the paper.

There is absolutely no build up of the hypotheses, they are just 'postulated' on page 9 and seem to appear out of thin air. Please include an argumentation/build up for each hypo by including a paragraph preceding the hypos in which the logic can be followed and we can see why (and in which direction) a hypothesis is drafted.

It is a bit odd that results of the analysis are included in introduction and dotted across the manuscript (e.g. p6: Nevertheless, this study does not provide any conclusive evidence on whether inventor networks in other technologies have small-world structures, and therefore further research is needed). This creates the impression that results were used to draft the text or, worse, guide the research. So, I would consider moving results to the latter half of the manuscript.

More profoundly, and taking into account all of the comments above, the impression is created that results were known before the hypotheses were drafted (HARKing)?

Context with regard to the evolution of the IPR system during the relevant time frame is lacking. For instance, there can be a number of reasons as to why inventor/patent networks grow (investments, government expenditures, subsidies for collaborative research, competition dynamics) and why inventors diversify (scientific advances, competition…) that go beyond pure ‘network laws”.

I have an issue with the data, based on the following statement with regard to the data used. Page 10: “The original dataset contains over 3.6 million patents ranging from 182,978 patents granted in 2007 to 392,618 in 2019.” From this, I take it that the authors are working with 'granted patents' and not with 'patent applications'? This approach can make for some crucial biases in the analyses. For instance, if ‘date granted’ is used to draft the networks per year (as is implied by the aforementioned statement, I find no reference to application date in the manuscript) then the authors should be aware that there are large differences between technology classes with regard to the time period between patent application and obtaining a granted patent. For instance: in ‘hot domains’ such as life sciences it can take up to 7 years to get to an approved patent while in other areas there are faster lead times (e.g. 3 years). Furthermore, the US government has created 'fast-tracks' for specific domains (e.g. clean tech under the Obama administration). This means that two patent applications done in the same year, let's take 2009, in two different technology classes will show up in different networks (e.g. one patent will be granted in 2012 and one in 2014). Wouldn’t this heavily bias the composition of your networks and the analysis done with regard to ‘technology classes’ and diversification? Furthermore, time between application and granting of a patent varies over time (see annual performance reports by USPTO indicating a decrease in the 'lag' over the past years).

Potential solution would be to construct the networks based on application date and redo the analyses. Either way, you will always be confronted with the fact that you are using only ‘granted patents’, which means you are working with a survivors dataset, so please highlight potential biases that might arise from this dataset?

P25: ‘we found that inventors patenting in fields better adjusted to network rules were also more specialized in nature.’ How sure are you about direction of this causality?

Reviewer #4: This paper attempted to analyze dynamic changes with static analysis with using USPTO data. I believe that this is an interesting paper and worthy of publication. I would also consider it appropriate for the journal content of PLOS ONE. However, I could not understand originality and value with insight of this paper. The minor revision is needed for these points. Also, some questions remain in the following points. I hope the author will improve the completeness of the document through revisions.

1. In this paper, the R package used for the analysis is described, but the detailed formulas and calculation process are not described. It is recommended to show the mathematical formulae for important analysis to understand the algorithm. Some, but not all, of the most important indicators in this study should be presented with mathematical formulas.

2. The reasons for the selection of the patent fields selected for this study should be stated. It is also recommended to state the characteristics of the field, even if it is a brief description. The reason is that the discussion in the concluding section will be insightful if it reflects these characteristics.

3. The weakest part of this study is that it only describes the results of adapting the indicators described in other papers. Of course, it is valuable in terms of reporting, but it is difficult to say that any new findings were obtained. Many of the hypotheses have already been stated in previous studies. I would like to know more about the originality of this study and the new facts derived from the analysis of this study.

4. Finally, in the results and discussion section, there is not enough discussion of why the results were determined the way they were. There is no clear view of some kind of theoretical background, so it is not possible to explain causality through the model. This may be out of the scope of this study, but the discussion is necessary to enhance the value of the study.

5. Overall, the review covers many studies, but I would say that the organization of the review is not structured. The review part should have a slightly more detailed section.

6. PLOS authors have the option to publish the peer review history of their article (what does this mean?). If published, this will include your full peer review and any attached files.

Reviewer #1: No

Reviewer #2: No

Reviewer #3: No

Reviewer #4: No

---

## [Author Response · Author response to Decision Letter 0]

30 Jun 2021

The authors would like to thank the Editor-in-Chief and the Referees for their time and effort in providing valuable comments and insights. We also appreciate the opportunity to improve our research and results. We agree with all the comments and we have revised our manuscript accordingly. The manuscript was significantly edited according to recommendations. The sections of Introduction and Literature review and hypotheses were extended to include previous studies, and explain the innovativeness of the present study. Statistics and figures in the new manuscript have all been updated based on the application date. The sections Results and Discussion were also restructured. To facilitate the work of the referees, we refer to the new manuscript by indicating the page number. We have included the Table 1 in the text (p. 16) as established in guidelines. Figures are also included separately as TIFF image files and at the end of this letter for your perusal. We made the data available directly to supporting information and online at https://github.com/amvallone/Data_PLOS_ONE_2020/tree/main. We hope that modifications done allowed for significant improvement of the text clarity.

 Response to comments from Anonymous Referee #1 

1. This study only states the importance of the topic in the Introduction section and fails to identify whether relevant studies have been proposed previously, determine the difference between the present study and previous studies, or explain the innovativeness of the present study.

We agree with the reviewer that we need to clarify the significance of the study and the research gaps. We have carefully revised the introduction and introduced major changes to the original text and research design. The paragraph has been revised as follows (p. 4): 

 “In this paper, we establish several hypotheses regarding the relationship between the structure and the dynamics of inventor networks, with a particular emphasis on technology domains. Following [1], the questions of interest we examine are: What do co-invention networks across technology domains look like, on a large scale? What sort of clustering behavior occurs? Do technology networks undergo a “phase transition", in which their behavior suddenly changes? What distributions and patterns do technology graphs maintain over time? To answer these questions, we study changes in the structural configurations of eight technology fields (IPC) via the power-law, small-world, preferential attachment, shrinking diameter, densification law, and ‘gelling point’ hypotheses.”

We also introduced the following text in the manuscript to briefly comment on this issue in the Literature review (p. 5):

“The network growth dynamics of technological fields have received less attention [2]. Historically, the core problem in studying innovations in different fields has been data availability, data silos and the limited access to cross-connections [3]. Today, the availability of large datasets that capture major activities in science and technology has created a revolution in the discipline of scientometrics [4]. One representative example is patent data produced by the USPTO, and particularly, the current organization of patent data in terms of IPC sections which provides the tools to address central questions of network analysis, such as those formulated in the study of different technological domains [5]. Many studies investigate technological and innovation activities by referring to links between nodes and the resulting networks [e.g., 6, 7-9]. However, in most, the metrics identified (at both node and network level) are mostly discussed from an economic perspective in only one field of technology or one geographic region. This offers us an opportunity to study the variations in the emerging patterns among collaboration networks, specifically inventor networks, through the lens of differently organized technological domains [5, 10-12]. 

2. In the literature review section, this paper only explains what is known about the static properties and dynamic properties. This study did not review those related to the influence of different statistical properties on the productivity of the invention. The study was needed to spell out the contribution.

In regards to the implications on the productivity, the referee raises a good point and we are aware of this limitation. However, the issue requires more attention and calculations. We hope to investigate this in a follow-up paper.

3. The development of the hypothesis requires sufficient theoretical or research support.

Following the reviewer’s suggestions, the following paragraphs have been changed/added at the Literature review and hypotheses section (p. 6 onwards):

“Evidence exists, however, showing slightly disturbed power law distributions [13], and also significant deviations from pure power-law distributions [14]. Furthermore, less is known about the inventor population dynamics within and across technology domains, and their repercussions on technology development [12]. With that in mind, our first related hypotheses based on SNA metrics on the growth dynamics of inventor networks in different technological domains is:

Hypothesis H1: the degree distribution for the co-inventor collaboration network, regardless of the technological domain, will follow a power-law, with γ>1”

“Researchers have found mixed evidence regarding the real small-worldliness of co-inventorship networks, which may suggest that the SW effect is either less pronounced or non-existent in patent collaborations [15-18]. This is due in part to the nature of the invention process, and the fact that search decisions for collaborator(s)/team members are not necessarily taken by the inventors themselves but rather the firm, i.e. the managers [19]. The literature suggests [10] that this may be due to the fragmentation of inventors´ networks and also the fact that said networks are small-worlds in some technology fields (e.g., Instruments, Industrial Processes, Chemicals & Materials and Pharmaceuticals & Biotechnology), while in other fields (e.g., Electrical Engineering & Electronics) they are not. Nevertheless, this study does not provide any conclusive evidence on whether inventor networks in other technologies have small-world structures, and therefore further research is needed. Therefore, our second hypothesis is:

Hypothesis H2: the co-inventor collaboration network, regardless of the technological domain, will exhibit properties typical of small-worlds, with small-coefficient σ>1.” 

“A recent study, however, found a negative association between the net change of the number of inventors and the net change of the number of links (that is, not all of the new inventors collaborate with inventors that are already established in the network), a result that appears to challenge the PA idea [20]. PA results have implications for the global innovation system in terms of organization, growth, and hierarchy, and they also present challenges in regard to the generation of new technologies [21]. These PA results therefore deserve further study in different fields, leading us to our third hypothesis. 

Hypothesis H3: the growth of the co-inventor collaboration network will be explained, regardless of the technological domain, by a sublinear PA regime, with α < 1.”

“First, graphs exhibit an average distance between nodes that often shrinks as part of their evolutionary process until it reaches an equilibrium [22]. Several experiments have been performed to verify that the shrinkage of diameters is not intrinsic to datasets, including the effects of missing past data [23] and disconnected components [24]. They seem to confirm the shrinkage as an inherent property of networks, and therefore we explore this in our dataset:

Hypothesis H4: the diameter of the co-inventor collaboration network, regardless of the technological domain, will keep shrinking over time with the addition of new edges until it reaches an equilibrium. 

Second, most real graphs evolve over time following the densification power-law or growth power-law, with the equation |E|(t)∝|N|(t)^β, and where |E|(t) and |N|(t) denote the number of edges and nodes of the graph at time step t with β being the densification exponent [23]. A densification exponent value greater than 1 indicates that the number of edges grows super-linearly with the number of nodes as the graph densifies over time. [24] recently describe similar increasing trends on the average degree over time and the number of edges as a function of the number of vertices in several collaboration networks (including patents, citations, and affiliation networks) indicating that graphs become dense. This could suggest that the densification of graphs is an intrinsic phenomenon. The question remains whether this property applies equally to all technology domain networks, and therefore we hypothesize:

Hypothesis H5: the co-inventor collaboration network, regardless of the technological domain, will densify over time with the addition of new nodes, with β>1.

Third, many small, disconnected components in real networks will merge over time and within a few periods form a giant connected component (GCC). This relationship is referred to as the ‘gelling point’ [1]. At this point, the GCC keeps growing, absorbing the vast majority of the newcomer nodes, while the network diameter continues to steadily decrease beyond this point. This observation was found to be consistent with the emergence of invisible colleges within scientific networks containing more than 50% of the nodes [25], and more recently, with the formation of a knowledge flow network of mobile inventors among US firms, institutions, and universities [26], the concentration of patents and citations around large corporations in Europe [27], as well as the dramatic aggregation in China of oligopolistic communities with key nodes taking central positions over time [18]. Additionally it seems to indicate a phase transition process in which small, isolated inventor networks end up forming one giant and connected component [15]. Therefore, the final hypothesis is:

Hypothesis H6: the co-inventor collaboration network, regardless of the technological domain, will exhibit a ‘gel point’, at which the diameter spikes and disconnected components gel into a giant component.”

4. The figure is unclear and vague, such as Figure 8.

All figures were revised to improve their quality. In particular, Fig 8 was replaced by a contingency table (Fig 7) that better reflects the inferences made in the article. Figures are included at the end of this cover letter for your revision.

5. You can also put all the figures and tables at the end of the paper to avoid breaking up the text.

Thanks, revised. We have included all the figures at the end of the paper. Table 1 (see p. 16) is included in the text as established in the guidelines. 

6. The discussion and conclusion of this study mostly explained the aforementioned the research hypothesis and did not draw relevant conclusions and management implications.

We have followed your recommendations and have carefully revised the Discussion and Conclusion section in the new manuscript. In particular, we would like to highlight the following findings (p. 26 onwards):

 We found evidence that technology field networks have a scale-free distribution, and exhibit SW properties. Thus, technology networks, regardless of the sector, are distributed in a highly skewed manner rather than following a normal pattern.

 “Our results also support the shrinking diameter and densification hypotheses for every technology sector, and we found partial evidence to support the gel point hypothesis, although some large and advanced fields’ networks such as Physics and Electricity do exhibit data consistent with this theory, as does the total network. This is expected in more mature networks since the number of nodes (inventors) remains constant over time, edges can only be added and not deleted, and densification naturally occurs [23]. This phenomenon is therefore consistent with critical transitions and the aggregation of isolated inventors as described in inventor network literature, and indicates that invention production, like any other knowledge production, undergoes a phase transition process during which small isolated inventor networks form one giant component [15].” 

 “The smallest technology groups, namely invention technologies in the fields of D. Textile and paper and E. Construction do not fit the gel point rules. Although there were indications of an early phase aggregation process, these inventor networks were still highly fragmented and much smaller in size than any of the other technology networks studied here, and this had important implications for their network topology.”

 “All technology sectors in the long run appear to comply with network laws. There also appear to be signs of institutional restrictions on collaboration and patenting. These restrictions appear to have reduced the magnitude of some of these rules, especially in smaller and newer fields.”

 “Overall, we found that the relative growth rates of diversification paths for inventors patenting in all sectors have also stabilized after a period of time. This may also be an indication that the inventor networks in all technology sectors are reaching a mature stage.” 

 “In summary, we show here that technology networks are complex and in the long run scale-invariant. This has important implications for the development of innovations over the different stages of technology networks.”

 Response to comments from Anonymous Referee #2 

The study of P.E. Pinto, G. Honores, and A. Vallone deals with statistical and dynamic properties of the collaboration network associated with US patents. The authors perform a thorough and encompassing characterization of the network properties such as degree distribution, betweenness centrality, network diameter, growth dynamics, etc. for several technological fields and analyze them. Moreover, they show what these network properties tell us about the inner working of the these technological fields and how innovations appear there. This aspect of their work shall be commended since network analysis per se is an established topic now, and the challenge now is in its application to the analysis of real, nonmathematical problems.

The paper is written clearly, pedagogically, and I like it. However, regarding presentation- Figs. 2,3,5,8 are technically unsatisfactory. I can't read the values on X- and Y-axes, the text and data resolution are insufficient.

Revised, thanks. Following the reviewer’s suggestions, all figures were revised and redone to improve their quality. We recommend to click image to download the figure the from the manuscript for improved quality. We have also included all the figures at the end of this letter.

With respect to data analysis: Fig.3 shows plots that serve to demonstrate the preferential attachment. I do not understand how the authors plotted the data - the figure caption is insufficient. What is the time window for these dynamic measurements? With respect to analysis, the data clearly can't be represented as dn/dt~n^(1+\\alpha) as the authors claim. The data for all fields seems to follow the linear dependence with offset, dn/dt~(n+n_0). Anyway, what is the meaning of this analysis with respect to the invention process? What does the linear or superlinear or sublinear trend tells us about inventor collaboration?

The figure was redone for clarity. The plot of data in Fig 3 shows the vertex degree k value in the horizontal axis, and the vertical axis shows the attachment function estimate (α). The data is estimated using the PAFit for PA of R [28].

According to the PA rule, only the linear case of α = 1 gives rise to a perfect PA function (that is, the chance that an inventor gets a new collaborator is proportional to their current number of collaborators). For 0 <α<1 the resulting degree distribution takes the form of a stretched exponential function, where for α > 1 a small group of large hubs receives all the new incoming connections or links in the network [29]. In other words, we can assume that popular inventors, i.e., those with more ties, will become even more popular in all of the networks. However, inventor popularity occurs at a sublinear growth rate in all networks we studied here. From this analysis, we can conclude that sub -linear PA governs the growth of inventor collaboration networks. That is, all the networks studied exhibit a tendency of PA (i.e., there is a higher probability to link to an inventor with a large number of connections), but with different scaling regimes and different exponents. We describe the following explanation in the text (p. 20): 

“In the sublinear PA regime, new nodes (i.e., inventors) are connected to old ones with a probability proportional to a fractional power of their degree. This asymptotic degree distribution is not a heavy-tailed but rather a stretched exponential degree distribution for sublinearly growing networks, which is characterized by an exponent smaller than one and a maximum degree that scales as a power of the logarithm of the number of nodes [30].”

"gelling point" (??) or "gel point"?

Following the literature [1], we use both “gel” or “gelling point” indistinctively to refer to the same phenomenon, the point at which the diameter spikes and disconnected components gel into a giant component. 

What do the studies of network diameter tell us- are there many more new inventors or just the circle of collaborators increases?

In the context of our paper, this is an indication of how effective (or ineffective) technology field networks are in connecting pairs of inventors in the largest component. As stated in the manuscript (pp. 21-22):

“In line with [31], this can be attributed to each inventor in these fields working with more inventors than is normal in other fields. The diameter of the overall largest component is 31; therefore, 31 steps separate those inventors furthest apart from each other in this network. The path length, defined as the average number of steps along the shortest paths for all possible pairs of the network, is just 8.4. This indicates the SW property, with indirect links between inventors becoming remarkedly short regardless of their location, industry or size of the network [26]. This is also consistent with our results showing overall decreasing values of diameter D and average path length L over time (D=78 and L=27.5 in 2002). Our results are in the range of those reported by other authors [10, 31].”

“A graphic representation of the absolute growth rate over time (Fig 4.b) shows size convergence among IPC sectors. The highlighted area in the graph represents the mean value of the growth rate per year plus/minus one standard deviation for all sectors over time. Both calculations, the zero-growth rate between technology sectors and the reduction of the variability rate over time, reinforce the idea of shrinking diameters followed by stabilization processes. Thus, based on our data, we conclude that the diameter of co-inventor collaboration networks shrinks over time in spite of the addition of new nodes, a phenomenon that can be observed in every technological domain although at different rates. Therefore, we accept hypothesis H4”

We then explain the reasons in the revised manuscript (p. 22):

“SNA studies have suggested that the shrinking diameter phenomenon can be attributed to the ‘densification’ [22] and the ‘gelling point’ [1] effects. To test the first effect and explore why we find differences among technology sections, we plot the number of nodes N(t) versus the number of edges E(t) in log-log scales over time and estimate the densification exponent β (Fig 5). All our graphs we studied densify over time with the number of edges growing superlinearly in the number of nodes, with β=1.15-2,28. This indicates that the number of edges E grows faster than the number of nodes N in these graphs. This relation is referred to as the densification power-law (DPL) or growth power-law [23]. The relatively good linear fit (〖 R〗^2~ 1) in a double logarithmic axis plot appears to also agree with the DPL, which indicates that the two variables are related through a power law of the type |E|(t)∝|N|(t)^β. It also explains away the shrinking diameter phenomenon observed in our real graphs [1].”

Shrinking or non-shrinking of the network diameter- what are the implications for the invention process? Or, any be these trends are the fingerprint of the new/old field?

The shrinking diameter is the fingerprint of a mature stage in a network. As stated in the revised manuscript (p. 27):

“Fourth, our results also support the shrinking diameter, and densification hypotheses for every technology sector, and found partial evidence to support the hypothesis of the gelling point, although some large and advanced fields’ networks such as Physics and Electricity do exhibit data consistent with this theory, as does the total network. This is expected in more mature networks the number of nodes (inventors) remains constant over time, edges can only be added and not deleted, and densification naturally occurs [23]. This phenomenon is therefore consistent with critical transitions and the aggregation of isolated inventors as described in inventor network literature, and indicates that invention production, like any other knowledge production, undergoes a phase transition process during which small isolated inventor networks form one giant component [15]. 

The smallest technology groups, namely invention technologies in the fields of D. Textile and paper and E. Construction does not fit the gelling point rules. Although there were indications of an early phase aggregation process, these inventor networks were still highly fragmented and much smaller in size than any of the other technology networks studied here, and this had important implications for their network topology.” 

I like the studies of diversification, but what do they tell us about dynamics of the patents in the field? Is new field less diverse than an old field? What does it tell us - the technological sector has reached a stationary state? Is it related to technological revolution? Does it tell us that the field recently passed through such revolution as does it tell us that the field stagnates?

We introduce ideas associated with the technological diversification of inventors to complement our network analysis. Our interest here is to search for indications of whether inventor networks in all technology sectors have reached a mature stage. We found interesting results which we describe in the revised Discussion and Conclusion section (p. 26):

“Finally, we explore a possible explanation by looking at the specialization/diversification patterns of inventors, and we found that inventors in all sectors over time have become more diversified. Overall, we found that the relative growth rates of diversification paths for inventors patenting in all sectors have also stabilized after a period of time. This may also be an indication that the inventor networks in all technology sectors are reaching a mature stage.”

In summary, the authors perform a thorough network analysis and draw sporadic conclusions about the maturity, organizational restraints, and mode of cooperation in different technological fields. To my mind, it would be nice to supplement this by the analysis of the invention process in each field as follows from network analysis. This does not require new measurements- just a section in discussion where the already existing material is rearranged differently - not according to network properties, but rather to each technological field.

We have followed your recommendations and have carefully revised the Discussion and Conclusion section in the new manuscript. In particular, we would like to highlight the following findings (p. 25 onwards):

 We found evidence that technology field networks have a scale-free distribution, and exhibit SW properties. Thus, technology networks, regardless of the sector, are distributed in a highly skewed manner rather than following a normal pattern.

 “Our results also support the shrinking diameter and densification hypotheses for every technology sector, and we found partial evidence to support the gel point hypothesis, although some large and advanced fields’ networks such as Physics and Electricity do exhibit data consistent with this theory, as does the total network. This is expected in more mature networks since the number of nodes (inventors) remains constant over time, edges can only be added and not deleted, and densification naturally occurs [23]. This phenomenon is therefore consistent with critical transitions and the aggregation of isolated inventors as described in inventor network literature, and indicates that invention production, like any other knowledge production, undergoes a phase transition process during which small isolated inventor networks form one giant component [15].” 

 “The smallest technology groups, namely invention technologies in the fields of D. Textile and paper and E. Construction do not fit the gel point rules. Although there were indications of an early phase aggregation process, these inventor networks were still highly fragmented and much smaller in size than any of the other technology networks studied here, and this had important implications for their network topology.”

 “All technology sectors in the long run appear to comply with network laws. There also appear to be signs of institutional restrictions on collaboration and patenting. These restrictions appear to have reduced the magnitude of some of these rules, especially in smaller and newer fields.”

 “Finally, we explore a possible explanation by looking at the specialization/diversification patterns of inventors, and we found that inventors in all sectors over time have become more diversified. Overall, we found that the relative growth rates of diversification paths for inventors patenting in all sectors have also stabilized after a period of time. This may also be an indication that the inventor networks in all technology sectors are reaching a mature stage. In a related issue, we found that “more productive” inventors tend to move more (perhaps in order to achieve a better match), and they were also more diversified. This may also suggest that inventors that move (assuming movement is voluntary) are also likely to be exposed to more i.e., diverse knowledge, hence the probability that they will invent more [32]. Whether or not this is permanent condition requires further analysis.”

 “In summary, we show here that technology networks are complex and in the long run scale-invariant. This has important implications for the development of innovations over the different stages of technology networks.”

 Response to comments from Anonymous Referee #3 

Reviewer #3: I would like to thank the authors for this very interesting paper and the rigorous way in which they conducted their analyses. I do have some issues with the paper, the main one being with the data used (see below). Besides that, I also make some recommendations to improve the paper and make it more convincing.

The paper is quite descriptive, describing the evolution of inventor networks over a given period. There is no clear research goal beyond describing the evolution of the network and as such it is unclear to the reader where the discussion about 'diversifying inventors' comes from and where it should be positioned in the literature. Is this an observation you stumbled upon or a discussion you want to engage in? In the latter case, you would have to include some theorization/hypotheses on this.

Furthermore, the authors mix theoretical concepts and measures/variables in the text (this starts at the introduction), I would strongly recommend to stick to 'concepts' in the first part of the paper and then move to the measures in the empirical part of the paper.

Yes, we agree with the reviewer. We introduce ideas associated with the technological diversification of inventors to complement our network analysis. Our interest here is to search for indications of whether inventor networks in all technology sectors have reached a mature stage. We found interesting results which we describe in the new result section. We hope that modifications done allowed for significant improvement of the text clarity.

 There is absolutely no build up of the hypotheses, they are just 'postulated' on page 9 and seem to appear out of thin air. Please include an argumentation/build up for each hypo by including a paragraph preceding the hypos in which the logic can be followed and we can see why (and in which direction) a hypothesis is drafted.

Following the reviewer’s suggestions, the following paragraphs have been changed/added at the Literature review and hypotheses section to provide the associated theoretical arguments preceding each hypothesis (p. 6 onwards):

“Evidence exists, however, showing slightly disturbed power law distributions [13], and also significant deviations from pure power-law distributions [14]. Furthermore, less is known about the inventor population dynamics within and across technology domains, and their repercussions on technology development [12]. With that in mind, our first related hypotheses based on SNA metrics on the growth dynamics of inventor networks in different technological domains is:

Hypothesis H1: the degree distribution for the co-inventor collaboration network, regardless of the technological domain, will follow a power-law, with γ>1”

“Researchers have found mixed evidence regarding the real small-worldliness of co-inventorship networks, which may suggest that the SW effect is either less pronounced or non-existent in patent collaborations [15-18]. This is due in part to the nature of the invention process, and the fact that search decisions for collaborator(s)/team members are not necessarily taken by the inventors themselves but rather the firm, i.e. the managers [19]. The literature suggests [10] that this may be due to the fragmentation of inventors´ networks and also the fact that said networks are small-worlds in some technology fields (e.g., Instruments, Industrial Processes, Chemicals & Materials and Pharmaceuticals & Biotechnology), while in other fields (e.g., Electrical Engineering & Electronics) they are not. Nevertheless, this study does not provide any conclusive evidence on whether inventor networks in other technologies have small-world structures, and therefore further research is needed. Therefore, our second hypothesis is:

Hypothesis H2: the co-inventor collaboration network, regardless of the technological domain, will exhibit properties typical of small-worlds, with small-coefficient σ>1.” 

“A recent study, however, found a negative association between the net change of the number of inventors and the net change of the number of links (that is, not all of the new inventors collaborate with inventors that are already established in the network), a result that appears to challenge the PA idea [20]. PA results have implications for the global innovation system in terms of organization, growth, and hierarchy, and they also present challenges in regard to the generation of new technologies [21]. These PA results therefore deserve further study in different fields, leading us to our third hypothesis. 

Hypothesis H3: the growth of the co-inventor collaboration network will be explained, regardless of the technological domain, by a sublinear PA regime, with α < 1.”

“First, graphs exhibit an average distance between nodes that often shrinks as part of their evolutionary process until it reaches an equilibrium [22]. Several experiments have been performed to verify that the shrinkage of diameters is not intrinsic to datasets, including the effects of missing past data [23] and disconnected components [24]. They seem to confirm the shrinkage as an inherent property of networks, and therefore we explore this in our dataset:

Hypothesis H4: the diameter of the co-inventor collaboration network, regardless of the technological domain, will keep shrinking over time with the addition of new edges until it reaches an equilibrium. 

Second, most real graphs evolve over time following the densification power-law or growth power-law, with the equation |E|(t)∝|N|(t)^β, and where |E|(t) and |N|(t) denote the number of edges and nodes of the graph at time step t with β being the densification exponent [23]. A densification exponent value greater than 1 indicates that the number of edges grows super-linearly with the number of nodes as the graph densifies over time. [24] recently describe similar increasing trends on the average degree over time and the number of edges as a function of the number of vertices in several collaboration networks (including patents, citations, and affiliation networks) indicating that graphs become dense. This could suggest that the densification of graphs is an intrinsic phenomenon. The question remains whether this property applies equally to all technology domain networks, and therefore we hypothesize:

Hypothesis H5: the co-inventor collaboration network, regardless of the technological domain, will densify over time with the addition of new nodes, with β>1.

Third, many small, disconnected components in real networks will merge over time and within a few periods form a giant connected component (GCC). This relationship is referred to as the ‘gelling point’ [1]. At this point, the GCC keeps growing, absorbing the vast majority of the newcomer nodes, while the network diameter continues to steadily decrease beyond this point. This observation was found to be consistent with the emergence of invisible colleges within scientific networks containing more than 50% of the nodes [25], and more recently, with the formation of a knowledge flow network of mobile inventors among US firms, institutions, and universities [26], the concentration of patents and citations around large corporations in Europe [27], as well as the dramatic aggregation in China of oligopolistic communities with key nodes taking central positions over time [18]. Additionally it seems to indicate a phase transition process in which small, isolated inventor networks end up forming one giant and connected component [15]. Therefore, the final hypothesis is:

Hypothesis H6: the co-inventor collaboration network, regardless of the technological domain, will exhibit a ‘gel point’, at which the diameter spikes and disconnected components gel into a giant component.”

It is a bit odd that results of the analysis are included in introduction and dotted across the manuscript (e.g. p6: Nevertheless, this study does not provide any conclusive evidence on whether inventor networks in other technologies have small-world structures, and therefore further research is needed). This creates the impression that results were used to draft the text or, worse, guide the research. So, I would consider moving results to the latter half of the manuscript.

More profoundly, and taking into account all of the comments above, the impression is created that results were known before the hypotheses were drafted (HARKing)?

The hypothesis used here are consistent with the aggregation of inventors over time as described in the inventor network literature. Generally speaking, the application of network theory to several complex systems has revealed that they share a number of common structural properties which facilitates their analysis and comparison [33], including power-law degree distribution [29], small-world [34], preferential attachment [35], and community structure [36]. More recently, studies have shown interest for the dynamic properties of such networks [1]. Using a novel database containing all patents granted by the USPTO, our study attempts to contribute to that stream of research in an area (technology networks) that has been less explored.

In addition, we have now provided the associated theoretical arguments preceding each hypothesis to complement the analysis. We have also redone all the statistic work based on the reviewers’ comments. 

In regards to SW, we have included the new results in Table 1 and the following text in the manuscript (p. 18):

“The small-coefficient σ is greater than 1 (σ=10,671; with SW=0.030). As such, the GCC has characteristics of SW, thus giving support for the hypothesis H1. We also determine the probability of randomly finding a network with a higher clustering coefficient for all sections. Our results consistently support the SW model of Watts and Strogatz over time and for all the technology fields (see Table 1 above)”

 

Table 1. Co-inventor network statistics by technological field (IPC) and total, 1999–2019

 Section Total network The largest component network

 A B C D E F G H 

Nodes (Vertices) 423,755 500,171 399,057 30,013 79,202 249,352 854,234 675,443 1,879,037 1,484,760

Edges (Links) 1,360,356 1,261,412 1,462,078 65,374 162,787 581,696 2,722,328 2,243,650 6,742,143 6,264,427

Degree centrality 6.4 5.0 7.3 4.4 4.1 4.7 6.4 6.6 7.2 8.4

Betweenness centrality 0.007 0.010 0.013 0.006 0.003 0.015 0.019 0.011 0.010 0.010

Clustering coefficient 0.415 0.437 0.379 0.614 0.520 0.467 0.299 0.267 0.229 0.225

k-core 4.60 3.78 5.05 3.65 3.35 3.55 4.30 4.33 4.57 5.18

Connected components

 46,523 

(261,177; 147) 58,251

(292,810; 195) 27,330

(295,309; 294) 5,135

(3,349; 1,965) 15,442 (19,065; 2,034) 34,433 (121,126; 233) 66,305 (629,482; 126) 46,254 (521,501; 53) 124,609 (1,484,760; 100) 1 

(1,484,760)

Largest component 

 Modularity 0.818 0.833 0.797 0.826 0.807 0.842 0.745 0.728 0.718 0.718

 Number of communities 13,044 15,974 14,407 255 1,217 6,960 32,088 25,964 65,944 65,944

 Average community size 20.02 18.33 20.50 13.13 15.66 17.40 19.62 20.08 22.51 22.51

 2-4 inventor/community 9% 8% 7% 9% 12% 9% 10% 10% 10% 10%

 5-9 24% 24% 24% 34% 28% 27% 23% 24% 22% 22%

 10 or more 67% 68% 69% 57% 60% 64% 67% 66% 68% 68%

 SW coefficient (σ) 20.65 121.60 144.10 26.99 44.65 103.59 124.74 113.89 10,671 10,671

Context with regard to the evolution of the IPR system during the relevant time frame is lacking. For instance, there can be a number of reasons as to why inventor/patent networks grow (investments, government expenditures, subsidies for collaborative research, competition dynamics) and why inventors diversify (scientific advances, competition…) that go beyond pure ‘network laws”.

Yes, the reviewer is correct. We added the following text (p. 28):

“The reasons why inventor networks grow (investments, subsidies for collaborative research, government expenditures) or why inventors diversify (scientific advances, competition dynamics) go beyond pure “network laws”. However, we find that in the long run that all types of technology networks appear to comply with these laws. There also appear to be signs of institutional restrictions on collaboration and patenting which appear to have reduced the size of the effects of some of these rules, especially in smaller and newer fields in the early periods of network formation. In summary, we show here that technology networks are complex and in the long run scale-invariant. This has important implications for the development of innovations over the different stages of technology networks.” 

Finally, we agree with the reviewer that we did not discuss the evolution of the IPR system in detail. The referee raises a good point and we are aware of this limitation. However, given the changes we introduced to the new version and the extension of the paper, we did not include IPR elements in the final version. 

I have an issue with the data, based on the following statement with regard to the data used. Page 10: “The original dataset contains over 3.6 million patents ranging from 182,978 patents granted in 2007 to 392,618 in 2019.” From this, I take it that the authors are working with 'granted patents' and not with 'patent applications'? This approach can make for some crucial biases in the analyses. For instance, if ‘date granted’ is used to draft the networks per year (as is implied by the aforementioned statement, I find no reference to application date in the manuscript) then the authors should be aware that there are large differences between technology classes with regard to the time period between patent application and obtaining a granted patent. For instance: in ‘hot domains’ such as life sciences it can take up to 7 years to get to an approved patent while in other areas there are faster lead times (e.g. 3 years). Furthermore, the US government has created 'fast-tracks' for specific domains (e.g. clean tech under the Obama administration). This means that two patent applications done in the same year, let's take 2009, in two different technology classes will show up in different networks (e.g. one patent will be granted in 2012 and one in 2014). Wouldn’t this heavily bias the composition of your networks and the analysis done with regard to ‘technology classes’ and diversification? Furthermore, time between application and granting of a patent varies over time (see annual performance reports by USPTO indicating a decrease in the 'lag' over the past years).

Potential solution would be to construct the networks based on application date and redo the analyses. Either way, you will always be confronted with the fact that you are using only ‘granted patents’, which means you are working with a survivors dataset, so please highlight potential biases that might arise from this dataset?

We greatly appreciate the reviewer for his/her meaningful and constructive comments on our manuscript. As the reviewer points out, there were errors in the form we originally estimated the networks per year. Following his/her comments, we have corrected and improved the revised manuscript. Statistics and graphs in the new manuscript have all been updated accordingly. This meant extensive work, which included rebuilding all collaborative networks based on the application date, and running all the analysis again. As briefly explained in the revised manuscript (pp. 11-12):

“The resulting dataset contained patents awarded by the USPTO between 2007 and 2019. However, they were filed over a longer period (1969-2019). This creates an additional problem: drawing annual networks based on the ‘granted’ date instead of the ‘filed’ date is inaccurate, since patent applications filed in one year but granted in another will end up showing up in different networks. To solve this assignation problem, we use the filed date to build the networks for each year. For graphical and calculation purposes, we removed patents filed before 1999 as they were insignificant in number and percentage (1,238 patents or 0.06% of the total). Thus, our final dataset contains 2,241,201 patents and 1,879,037 inventors. Note that a patent may contain several technical objects and consequently be assigned to more than one section (in our case, 574,737 patents). These patents are counted in order to map each technological domain. In the final dataset, nearly half of (47.9%) inventors patented only once, but 96.2% of patents were made by repeat inventors with more than one patent. We could thus track diversification as repeat inventors patented in more than one domain over time.” 

We also acknowledge the survivor bias in the limitations of this study in the new manuscript (p. 29): 

“The comparison with other studies is also limited by the fact that we use patent grants (rather than applications), which could have caused an overestimation of team size as single inventors are less likely to obtain a patent, or an underestimation an because not all inventors are listed on a patent application [37]. There are also large differences between technology classes with regard to the time period between applying and obtaining a granted patent as described in the text. Future research could take into consideration lag periods to account for a comparison between nascent and mature technology domains.” 

P25: ‘we found that inventors patenting in fields better adjusted to network rules were also more specialized in nature.’ How sure are you about direction of this causality?

Yes, the reviewer is right. We did not intent to test causality here. We apologize for any misunderstanding that the original wording created. Our work found supporting evidence to three related issues:

First, we found that inventors in all sectors over time have become more diversified. The changes in the diversification levels are statistically significant.

Second, the relative growth rates of diversification paths for inventors patenting in all sectors have also stabilized after a period of time. This may also be an indication that the inventor networks in all technology sectors are reaching a mature stage.

Third, larger sectors experienced higher specialization levels and a more focused diversification.

 Response to comments from Anonymous Referee #4 

1. In this paper, the R package used for the analysis is described, but the detailed formulas and calculation process are not described. It is recommended to show the mathematical formulae for important analysis to understand the algorithm. Some, but not all, of the most important indicators in this study should be presented with mathematical formulas.

Thanks, revised. The following algorithms were used:

 SW package of R “brainwaver” [38].

 PAFit for PA of R [28].

 PowerRlaw R package [39].

 Community detection [36] using igraph R package [40].

The parameter testing was performed using R. The formulas were included in the revised manuscript (pp. 12-14):

H1: The discrete mass function of a power law distribution is:

P(X=k)=k^(-γ)/(ξ(γ,k_min))

Where ξ(γ,k_min) is the is the generalized zeta function [41]. The maximum likelihood estimator for the γ parameter is:

γ ^≃1+n[∑_(i=1)^n▒〖ln k_i/(k_min-0.5)〗]^(-1)

The estimation procedure and the algorithms are described in [39].

H2: The network small-worldness is quantified by the σ coefficient, calculated by comparing clustering (C) and path length (L) of a given network to an equivalent random network with the same degree [42]:

σ=(C⁄C_random )/(L⁄L_random )

When σ>1; C>C_random and L≈L_random, the network is small-world. To estimate the σ we used the algorithms available in the “small.world” function of the “brainwaver” R package [38].

H3: Following [43] in the PA mechanism, the probability P_i (t) that a node v_i acquires a new edge at time t is proportional to a positive function, A_(k_i ) (t), of its current degree k_i (t). The function A_k is the attachment function which assumes a log-linear form of k^α, with α is the attachment exponent. In the fitness mechanism the probability P_i (t) that a node v_i acquires a new edge depends only on the positive number η_i that can be interpreted as the intrinsic attractiveness. In their combined form, the probability P_i (t) is proportional to the product of A_(k_i ) (t) and η_i : 

P_i (t)∝ k^α×η_i

We follow the estimated parameter α, algorithms and procedures described in [43]

H4 The diameter of a network is the longest of all the calculated shortest paths in a network. The diameter was estimated with the available algorithms in the “diameter” function of the igraph R package [40].

H5 The densification power-law or growth power-law can be expressed as:

|E|(t)∝|N|(t)^β

where |E|(t) is the number of edges of a graph at time step t, |N|(t) denote the number of nodes of the graph at time step t and β is the densification exponent [23]. Using ordinary least square, the β parameter was estimated using the following model specification:

log⁡(|E|(t))=δ+β log(|N|(t))+u

H6 The gel point was estimated as described in [1].

2. The reasons for the selection of the patent fields selected for this study should be stated. It is also recommended to state the characteristics of the field, even if it is a brief description. The reason is that the discussion in the concluding section will be insightful if it reflects these characteristics.

Patents were classified according to the International Patent Classification (IPC) model. The classification code attached to a patent defines the technological class of the patent [44]. We added the names of the eight major sections in the Introduction (p. 4): 

“The IPC divides patentable technology into eight major sections: A: Human necessities, B: Performing operations; Transporting, C: Chemistry; Metallurgy, D: Textiles; Paper, E: Fixed constructions, F: Mechanical engineering; Lighting; Heating; Weapons; Blasting, G: Physics, and H: Electricity.” 

We have also stated the reasons for the selection of the technological fields as follows (pp. 4-5):

“The assumptions underpinning our analysis are that each IPC category of patents represents the space of a specific technology field (as a network), and that the division between categories enables us to not only identify the characteristic structure of each technology field network [11], but also to map the inventor relationship data across different technology domains [12].”.

“The network growth dynamics of technological fields have received less attention [2]. Historically, the core problem in studying innovations in different fields has been data availability, data silos and the limited access to cross-connections [3]. Today, the availability of large datasets that capture major activities in science and technology has created a revolution in the discipline of scientometrics [4]. One representative example is patent data produced by the USPTO, and particularly, the current organization of patent data in terms of IPC sections which provides the tools to address central questions of network analysis such as those derived in the study of different technological domains [5]. Many studies investigate technological and innovation activities by referring to links between nodes and the resulting networks [e.g., 6, 7-9]. However, the metrics identified (at both node and network level) are mostly discussed from an economic perspective in only one field of technology or one geographic region. This offers us an opportunity to study the variations in the emerging patterns among collaboration networks, specifically inventor networks, through the lens of differently organized technological domains [5, 10-12].” 

3. The weakest part of this study is that it only describes the results of adapting the indicators described in other papers. Of course, it is valuable in terms of reporting, but it is difficult to say that any new findings were obtained. Many of the hypotheses have already been stated in previous studies. I would like to know more about the originality of this study and the new facts derived from the analysis of this study.

Our contribution resides in the use of network analyses to examine the dynamic behavior of eight major sections and in the introduction of the diversification analysis. The hypotheses have been derived from the literature. However, to the best of the authors' knowledge, there have been no such studies of technology sectors, and the metrics identified (at both node and network level) have been mostly discussed from an economic perspective in only one field of technology or one geographic region. We also use static analysis to reveal how different collaboration networks are structured. The following changes have been applied in the new manuscript to make our results more robust (pp. 24-25):

“Recent empirical studies have described the explanatory power of the diversification paths of inventors and inventor organizations on technology field network maps [2]. Therefore, one might speculate that the compliance/non-compliance with a network’s rules in some of the real graphs observed here may be related to the inventor diversification pattern, among other factors. Using a contingency table (Fig. 7a), it is possible to represent the observed frequencies of the diversification components of each sector. In general, tables made up of R rows and C columns are considered. For i=1,..,R and j=1,..,C, p_ij represents the probability that a random observation belonging to a population under study will be classified in the ith row and jth column of the table. Denoted by p_(i•) is the marginal probability that an observation will be classified in the ith row of the table. Similarly, p_(•j) denotes the marginal probability that an observation will be classified in the jth column of the table. The sum of the probabilities of all the cells in the contingency table must add up to 1. To analyze dynamic changes of each sector from 1999 to 2019, we also split data into four equivalent periods (data for the year 1999 was included in the first period). For explanation purposes, we use the Pr index to rate diversification. Here, it corresponds to one minus the main diagonal value of the contingency table for each sector.

The static view (i.e., the average probability values over the total period) shows that the most specialized inventors correspond to those who have patented in A (Pr = 0.432), G (Pr = 0.403) and H (Pr = 0.415), with these two last sections also complementing each other relatively well (see highlighted areas). Sections A, B, C, D, E, and F have all average Pr-values between 0.482 and 0.490, whereas section D displays the highest diversification level, i.e., Pr>0.5. This indicates that at least 1 out of every 2 patents registered in section D is also registered in another one. 

The more dynamic view indicates an increase in the diversification levels over time in all sectors, including in those sectors which were highly specialized at first (e.g., sectors E, G, and H). Inventors in sectors B, C, D, E, and F have also become more diversified than specialized surpassing the threshold Pr>0.5. They also appear to be patenting in various sectors at the same time.

While the analysis of the contingency tables shows a growing trend towards diversification, differences between periods can be considered subtle. To evaluate how likely it is that any observed difference between the three periods arose by chance using a Pearson's chi-squared test (χ^2). The null hypothesis was rejected at a significance level of 1 per cent. Therefore, our study confirms that in the long run changes in diversification levels are statistically significant and they grow over time.

The contingency table also allows us to measure relationships between sectors. When analyzing cells other than the main diagonal ones, the i,j values show the proportion of inventors who have jointly registered a patent. We can observe the set of relationships intensifies over time in some technological sectors. For instance, in the first observed period (1999-2004), 19.7% of inventors who registered patents in sector H also did so in G, and 14.6% of inventors who registered in G did so at the same time in H. In the last period (2015-2019), these numbers increased to 27.0%, and 24.4%, respectively. A similar relationship increasing over time is found with inventors patenting simultaneously in F and B, but with a lower intensity (a change from 12.1% to 17.2%). It is also possible to notice sectors with a stable relationship over time (e.g., between sectors D and B, with 17.4% of joint patents). Finally, there are also relationships that have lost intensity over time (e.g., A and C), although they remain relevant. Other sectors have few or almost no relationships (e.g., A, G, and H with D). 

We observe the largest sectors (H, G, and B) showing an inflow greater than the smallest sectors (e.g., D and E). The data also allows us to detect a set of increasing relationships in the type, as is the case between G and H whose relationship increases period by period, and the case of section B, which is becoming more relevant to G and H. This reinforces the idea of both higher specialization levels and more focused diversification in the larger sectors.

* Note: A Pearson's chi-squared test (χ^2) was applied:

χ^2=∑_i▒∑_j▒(S_(i,j)^(t+1)-S_(i,j)^t)/(S_(i,j)^t )

Where S_(i,j)^(t+1) are the values in the final period and S_(i,j)^t are the values in the initial period for a matrix S.

Also, in the Discussion and Conclusion section we have included the following text (p. 28):

“Finally, we explore a possible explanation by looking at the specialization/diversification patterns of inventors, and we found that inventors in all sectors over time have become more diversified. Overall, we found that the relative growth rates of diversification paths for inventors patenting in all sectors have also stabilized after a period of time. This may also be an indication that the inventor networks in all technology sectors are reaching a mature stage.”

4. Finally, in the results and discussion section, there is not enough discussion of why the results were determined the way they were. There is no clear view of some kind of theoretical background, so it is not possible to explain causality through the model. This may be out of the scope of this study, but the discussion is necessary to enhance the value of the study.

We have revised the text to include implications. For the revised manuscript, the main findings of this study are summarized below (pp. 26-27):

 We found evidence that technology field networks have a scale-free distribution, and exhibit SW properties. Thus, technology networks, regardless of the sector, are distributed in a highly skewed manner rather than following a normal pattern.

 “our results also support the shrinking diameter and densification hypotheses for every technology sector, and we found partial evidence to support the gel point hypothesis, although some large and advanced fields’ networks such as Physics and Electricity do exhibit data consistent with this theory, as does the total network. This is expected in more mature networks since the number of nodes (inventors) remains constant over time, edges can only be added and not deleted, and densification naturally occurs [23]. This phenomenon is therefore consistent with critical transitions and the aggregation of isolated inventors as described in inventor network literature, and indicates that invention production, like any other knowledge production, undergoes a phase transition process during which small isolated inventor networks form one giant component [15].” 

 “The smallest technology groups, namely invention technologies in the fields of D. Textile and paper and E. Construction do not fit the gel point rules. Although there were indications of an early phase aggregation process, these inventor networks were still highly fragmented and much smaller in size than any of the other technology networks studied here, and this had important implications for their network topology.”

 “All technology sectors in the long run appear to comply with network laws. There also appear to be signs of institutional restrictions on collaboration and patenting. These restrictions appear to have reduced the magnitude of some of these rules, especially in smaller and newer fields.”

 “The relative growth rates of diversification paths for inventors patenting in all sectors have also stabilized after a period of time. This may also be an indication that the inventor networks in all technology sectors are reaching a mature stage.”

5. Overall, the review covers many studies, but I would say that the organization of the review is not structured. The review part should have a slightly more detailed section.

 Yes, we agree with the reviewer about the structure of the paper. The manuscript was significantly edited according to recommendations. The sections of Introduction and Literature review and hypotheses were extended to include previous studies. The sections Results and Discussion were also restructured. All figures were revised and redone. 

We hope that with the aforementioned changes we have addressed all issues mentioned in the reviewers’ comments. 

Sincerely, 

The authors. 

References

1. McGlohon M, Akoglu L, Faloutsos C. Statistical Properties of Social Networks. In: Aggarwal CC, editor. Social network data analytics: Springer; 2011. p. 17-42.

2. Yan B, Luo J. Measuring technological distance for patent mapping. Journal of the Association for Information Science and Technology. 2017;68(2):423-37. doi: https://doi.org/10.1002/asi.23664.

3. Griliches Z. Productivity, R&D, and the Data Constraint. The American Economic Review. 1994;84(1):1-23.

4. Zeng A, Shen Z, Zhou J, Wu J, Fan Y, Wang Y, et al. The science of science: From the perspective of complex systems. Physics Reports. 2017;714-715:1-73. doi: https://doi.org/10.1016/j.physrep.2017.10.001.

5. Leydesdorff L, Kushnir D, Rafols I. Interactive overlay maps for US patent (USPTO) data based on International Patent Classification (IPC). Scientometrics. 2014;98(3):1583-99.

6. Marra A, Antonelli P, Pozzi C. Emerging green-tech specializations and clusters – A network analysis on technological innovation at the metropolitan level. Renewable and Sustainable Energy Reviews. 2017;67:1037-46. doi: https://doi.org/10.1016/j.rser.2016.09.086.

7. Tóth G, Lengyel B. Inter-firm inventor mobility and the role of co-inventor networks in producing high-impact innovation. The Journal of Technology Transfer. 2019. doi: 10.1007/s10961-019-09758-5.

8. Guan J, Liu N. Exploitative and exploratory innovations in knowledge network and collaboration network: A patent analysis in the technological field of nano-energy. Research Policy. 2016;45(1):97-112. doi: https://doi.org/10.1016/j.respol.2015.08.002.

9. Chai K-C, Yang Y, Sui Z, Chang K-C. Determinants of highly-cited green patents: The perspective of network characteristics. PloS One. 2020;15(10):e0240679. doi: 10.1371/journal.pone.0240679.

10. Lissoni F, Llerena P, Sanditov B. Small worlds in networks of inventors and the role of academics: An analysis of France. Industry and Innovation. 2013;20(3):195-220. doi: https://doi.org/10.1080/13662716.2013.791128.

11. Huang H-C, Su H-N. The innovative fulcrums of technological interdisciplinarity: An analysis of technology fields in patents. Technovation. 2019;84-85:59-70. doi: https://doi.org/10.1016/j.technovation.2018.12.003.

12. Alstott J, Triulzi G, Yan B, Luo J. Inventors’ explorations across technology domains. Design Science. 2017;3(e20). doi: 10.1017/dsj.2017.21.

13. Ferrara M, Mavilia R, Pansera BA. Extracting knowledge patterns with a social network analysis approach: an alternative methodology for assessing the impact of power inventors. Scientometrics. 2017;113(3):1593-625. doi: 10.1007/s11192-017-2536-2.

14. Wagner CS, Leydesdorff L. Network structure, self-organization, and the growth of international collaboration in science. Research Policy. 2005;34(10):1608-18.

15. Fleming L, King C, Juda A. Small worlds and regional innovation. Organization Science. 2007;18(6):938-54.

16. Chen Z, Guan J. The impact of small world on innovation: An empirical study of 16 countries. Journal of Informetrics. 2010;4(1):97-106. doi: https://doi.org/10.1016/j.joi.2009.09.003.

17. Ebadi A, Schiffauerova A. On the relation between the small world structure and scientific activities. PloS One. 2015;10(3):e0121129.

18. Ye Y, De Moortel K, Crispeels T. Network dynamics of Chinese university knowledge transfer. The Journal of Technology Transfer. 2020;45:1228–54. doi: https://doi.org/10.1007/s10961-019-09748-7.

19. Crescenzi R, Nathan M, Rodríguez-Pose A. Do inventors talk to strangers? On proximity and collaborative knowledge creation. Research Policy. 2016;45(1):177-94.

20. Fritsch M, Zoellner M. The fluidity of inventor networks. The Journal of Technology Transfer. 2019. doi: 10.1007/s10961-019-09726-z.

21. Ribeiro LC, Rapini MS, Silva LA, Albuquerque EM. Growth patterns of the network of international collaboration in science. Scientometrics. 2018;114(1):159-79. doi: 10.1007/s11192-017-2573-x.

22. Leskovec J, Kleinberg J, Faloutsos C, editors. Graphs over time: densification laws, shrinking diameters and possible explanations. Proceedings of the eleventh ACM SIGKDD international conference on Knowledge discovery in data mining; 2005.

23. Leskovec J, Kleinberg J, Faloutsos C. Graph evolution: Densification and shrinking diameters. ACM Trans Knowl Discov Data. 2007;1(1):2–es. doi: 10.1145/1217299.1217301.

24. Raj PM, Mohan A, Srinivasa KG. Power Law. Practical Social Network Analysis with Python Computer Communications and Networks. Cham.: Springer; 2018.

25. Guimera R, Uzzi B, Spiro J, Amaral LAN. Team assembly mechanisms determine collaboration network structure and team performance. Science. 2005;308(5722):697-702. doi: 10.1126/science.1106340.

26. Kiss IM, Buzás N. Communities and central nodes in the mobility network of US inventors. Journal of Innovation Management. 2015;3(4):96-118.

27. Chakraborty M, Byshkin M, Crestani F. Patent citation network analysis: A perspective from descriptive statistics and ERGMs. PloS One. 2020;15(12):e0241797. doi: 10.1371/journal.pone.0241797.

28. Pham T, Sheridan P, Shimodaira H. PAFit: A statistical method for measuring preferential attachment in temporal complex networks. PLoS ONE. 2015;10(9):e0137796.

29. Newman ME. The Structure and Function of Complex Networks. SIAM Review. 2003;45(2):167 - 256. PubMed PMID: doi:10.1137/S003614450342480.

30. Jog V, Loh P. Analysis of Centrality in Sublinear Preferential Attachment Trees via the Crump-Mode-Jagers Branching Process. IEEE Transactions on Network Science and Engineering. 2017;4(1):1-12. doi: 10.1109/TNSE.2016.2622923.

31. Balconi M, Breschi S, Lissoni F. Networks of inventors and the role of academia: an exploration of Italian patent data. Research Policy. 2004;33(1):127-45.

32. Trajtenberg M, Shiff G, Melamed R. The" names game": Harnessing inventors' patent data for economic research. In: Research NBoE, editor.: National Bureau of Economic Research; 2006.

33. Barabási A-L, Bonabeau E. Scale-free networks. Scientific American. 2003;(May):50-9.

34. Watts DJ, Strogatz SH. Collective dynamics of ‘small-world’ networks. Nature. 1998;393(6684):440-2.

35. Barabási A-L, Albert R. Emergence of Scaling in Random Networks. Science. 1999;286(5439):509-12. doi: 10.1126/science.286.5439.509 %J Science.

36. Girvan M, Newman MEJ. Community structure in social and biological networks. Proceedings of the National Academy of Sciences. 2002;99(12):7821-6. doi: 10.1073/pnas.122653799.

37. Pinto PE, Vallone A, Honores G. The structure of collaboration networks: Findings from three decades of co-invention patents in Chile. Journal of Informetrics. 2019;13(4):100984. doi: https://doi.org/10.1016/j.joi.2019.100984.

38. Achard S. brainwaver: Basic wavelet analysis of multivariate time series with a visualisation and parametrisation using graph theory. R package version. 2012;1.

39. Gillespie CS. Fitting Heavy Tailed Distributions: The poweRlaw Package. Journal of Statistical Software. 2015;64(2):1-16. doi: http://www.jstatsoft.org/v64/i02/.

40. Csardi G, Nepusz T. The igraph software package for complex network research. InterJournal Complex Systems. 2006;1695(5):1-9.

41. Abramowitz M, Stegun IA. Handbook of mathematical functions with formulas, graphs, and mathematical tables: US Government printing office; 1964.

42. Humphries MD, Gurney K. Network ‘Small-World-Ness’: A Quantitative Method for Determining Canonical Network Equivalence. PloS One. 2008;3(4):e0002051. doi: 10.1371/journal.pone.0002051.

43. Pham T, Sheridan P, Shimodaira H. PAFit: An R Package for the Non-Parametric Estimation of Preferential Attachment and Node Fitness in Temporal Complex Networks. 2020. 2020;92(3):30. Epub 2020-02-18. doi: 10.18637/jss.v092.i03.

44. Noruzi A, Abdekhoda M. Mapping Iranian patents based on International Patent Classification (IPC), from 1976 to 2011. Scientometrics. 2012;93(3):847-56.

---

## [Decision Letter · Decision Letter 1]

22 Jul 2021

PONE-D-20-41087R1

Exploring the topology and dynamic growth properties of co-invention networks and technology fields

PLOS ONE

Dear Dr. PINTO,

Thank you for submitting your manuscript to PLOS ONE. After careful consideration, we feel that it has merit but does not fully meet PLOS ONE’s publication criteria as it currently stands. Therefore, we invite you to submit a revised version of the manuscript that addresses the points raised during the review process.

We look forward to receiving your revised manuscript.

Kind regards,

Petter Holme, Ph.D.

Academic Editor

PLOS ONE

Journal Requirements:

Reviewers' comments:

Reviewer's Responses to Questions

**Comments to the Author**

1. If the authors have adequately addressed your comments raised in a previous round of review and you feel that this manuscript is now acceptable for publication, you may indicate that here to bypass the “Comments to the Author” section, enter your conflict of interest statement in the “Confidential to Editor” section, and submit your "Accept" recommendation.

Reviewer #2: (No Response)

Reviewer #4: All comments have been addressed

2. Is the manuscript technically sound, and do the data support the conclusions?

Reviewer #2: Partly

Reviewer #4: Yes

3. Has the statistical analysis been performed appropriately and rigorously? 

Reviewer #2: Yes

Reviewer #4: Yes

4. Have the authors made all data underlying the findings in their manuscript fully available?

Reviewer #2: No

Reviewer #4: Yes

5. Is the manuscript presented in an intelligible fashion and written in standard English?

Reviewer #2: No

Reviewer #4: Yes

6. Review Comments to the Author

Reviewer #2: The authors made a great effort to answer the comments of the four reviewers and managed to improve their manuscript. Although I do not agree with some of the statements, I like this work and want it to be published. However, the Figures are unaceeptable. First of all, the drawings are unintelligible and shall be redrawn, the font size shall be increased. Fig. 4 misses vertical axis, Fig. 6 misses plot legend, the plot legend of the Fig. 4 is outside the figure area. The most important problem is with the Figure captions. They are simply absent. This manuscript mixes Figure captions and Figure description/Discussion together. This is not good. The Figure caption shall be separate and tell to the readers what we see in the Figure, what is what. The reader shall understand the paper in grosso modo through looking at the figures and reading figure captions.

Fig. 1 - I do not understand anything here. There are no axes, no titles, no legend.

Fig.2 shows a fat-tail distribution of the number of the authors of the papent. I do not understand the importance of this indicator for the invention process. Do you mean that the large number of collaborators submitting a patent indicates a collective nature of the invention process?

Fig. 3- With respect to the preferential attachment - I do not know what are the implications of the validity of this hypothesis with respect to the assessment of the invention process, but Fig. 3 clearly disqualifies this hypothesis.

Reviewer #4: Thank you for your efforts in revision. The third and fourth points in the conclusion section were very meaningful. I think the value of the paper would be further enhanced if these (third and fourth points) were emphasized and stated. The first and second findings are relatively common, so I honestly don't think they are very important.

7. PLOS authors have the option to publish the peer review history of their article (what does this mean?). If published, this will include your full peer review and any attached files.

Reviewer #2: No

Reviewer #4: No

---

## [Author Response · Author response to Decision Letter 1]

14 Aug 2021

The authors would like to thank the Editor-in-Chief and the Referees for their time and effort in providing valuable comments and insights. We also appreciate the opportunity to improve our research and results. We agree with all the comments and we have revised our manuscript accordingly. We hope that modifications done allowed for significant improvement of the text clarity.

 Response to comments from Anonymous Reviewer #2 

Reviewer #2: The authors made a great effort to answer the comments of the four reviewers and managed to improve their manuscript. Although I do not agree with some of the statements, I like this work and want it to be published. However, the Figures are unacceptable. First of all, the drawings are unintelligible and shall be redrawn, the font size shall be increased. Fig. 4 misses vertical axis, Fig. 6 misses plot legend, the plot legend of the Fig. 4 is outside the figure area. The most important problem is with the Figure captions. They are simply absent. This manuscript mixes Figure captions and Figure description/Discussion together. This is not good. The Figure caption shall be separate and tell to the readers what we see in the Figure, what is what. The reader shall understand the paper in grosso modo through looking at the figures and reading figure captions.

Thanks for the comments. We agree that poor-quality images are unacceptable. In order to create high-quality images, 

in this revised version of the paper, Figures are included separately as EDS image files as established in the figure preparation guidelines. We also ensure that all the images have a resolution of at least 300 pixels per inch and appear sharp, not pixelated. 

Fig. 1 - I do not understand anything here. There are no axes, no titles, no legend.

A: We agree with the reviewer that we need to clarify Fig. 1. The paragraph has been revised as follows (pp. 17-18):

“Fig 1 visualizes the layout for the total and IPC technology networks using Cytoscape's algorithm [1]. The Cytoscape software is designed to visualize very large networks [2]. The central organizing principle of Cytoscape is a network graph, with inventors represented as nodes and interactions represented as links or edges between nodes. The network core refers to a central and densely connected set of network nodes (the largest connected component of the network), while the periphery of the network denotes the sparsely connected set of nodes, which are linked to the core. The non-connected or isolated nodes from the bulk of the network are shown to the side and bottom of the central figure” 

For representation, network graphs are presented without axes. However, to facilitate the interpretation we included a brief text pointing to the core, periphery, and the non-connected set of nodes. We also improve the quality and color of the image submission, and include a new title: Visualization of the technology co-inventor networks using Cytoscape, 1999–2019. 

Fig.2 shows a fat-tail distribution of the number of the authors of the paper. I do not understand the importance of this indicator for the invention process. Do you mean that the large number of collaborators submitting a patent indicates a collective nature of the invention process?

A: Fig. 2 presents a histogram of the distribution of inventors according to the number of connections that they have. Data for 1999, 2000 and 2019 are plotted to investigate the inter-temporal stability of this network [3]. In particular, Fig. 2 presents for 2019 the distribution of the 1.8 million vertices (inventors) according to the number of connections (degree) that they have presented (6.7 million edges or links in total). For the total network in 2019, there are 267.626 inventors with one link, and one inventor with 1.346 links. The power law (the straight line in a log x log scale) behavior of the distribution of connections (links) indicates that the co-inventorship collaboration network is a scale-free one. Power-law-like distributions for degrees state that there exist many low degree nodes (a large number of nodes with not so many connections), whereas only a few high degree nodes (a small number of nodes with a very high connection) in real graphs. [4]. However, as discussed in the paper (pp.19-20): 

“An interesting fact is that some of the graphs will exhibit power-law properties over the years, whereas others will not. For instance, the total network does not exhibit a scale-free range in the degree distribution in 1999. This can be explained because our empirical data contain only those nodes and links that have been created that year. By 2009 (as displayed in Fig 2), the distribution is already power-law (γ=4.37) and will stay the same in subsequent years. Section C is the only network that exhibits power-law distributions in both 2009 and 2019. Section A shows power-law behavior in 2009 but not in 1999 or 2019. Compared to the other distributions (and with an average confidence level of 99%), sectors A y H behave like an exponential, sectors D and E behave like a Poisson, and sector G behaves like a lognormal model. Therefore, and based on our data, we found indications of scale-free networks, but also significant deviations from an ideal power-law in some of these networks over time. Thus, the hypothesis H2, which states that the co-inventor collaboration network will follow a power-law regardless of the technological domain, is not supported by our analysis.”

Fig. 3- With respect to the preferential attachment - I do not know what are the implications of the validity of this hypothesis with respect to the assessment of the invention process, but Fig. 3 clearly disqualifies this hypothesis.

A: Fig 3 shows the vertex degree k value in the horizontal axis, and the attachment function estimate (α) in the vertical axis. The reviewer is correct: only the case of α = 1 gives rise to a perfect PA function. For 0 <α<1 the resulting degree distribution exhibits sublinear PA, whereas for α > 1 superlinear attachment [5]. The paragraph has been revised as follows (pp. 20-21):

“Our measurements using aggregated data from the category field networks show that the growth of co-inventorships can be explained based on the organizing principle of PA, although the attachment mechanism deviates from an ideal power-law. Applying SNA tools, our data reveals a sublinear preferential attachment model with exponent α=0.67-0.72. In the sublinear PA regime, new nodes (i.e., inventors) are connected to old ones with a probability proportional to a fractional power of their degree. In this case, the network becomes a gel-like where every node is connected to all other nodes and degree distribution is the stretched exponential rather than power-law [6]. This asymptotic degree distribution for sublinearly growing networks is characterized by an exponent smaller than one and a maximum degree that scales as a power of the logarithm of the number of nodes [7]. Therefore, our results from using large datasets support Hypothesis H3, that is, the sublinear regime, and this is consistent with previous measurements on scientific collaboration [6, 8]. This also indicates that a perfect PA function (that is, the chance that an inventor gets a new collaborator is proportional to their current number of collaborators) is not supported in our study.”

We also introduced the following text in the manuscript to briefly comment on this issue (pp. 26-27):

“We can infer that the observed distributions fit in a growth model in which the source of added edges are chosen according to a sublinear PA, but the destination is selected at random [9]. This is indicative of an inverse Matthew effect or a cumulative disadvantage and implies that past patent activity is not necessarily correlated with whatever growth mechanism is actually at play [10]. Therefore, we cannot assume that there is a clear growth mechanism for making new connections for inventors with a high degree [9].”

Response to comments from Anonymous Reviewer #4

Reviewer #4: Thank you for your efforts in revision. The third and fourth points in the conclusion section were very meaningful. I think the value of the paper would be further enhanced if these (third and fourth points) were emphasized and stated. The first and second findings are relatively common, so I honestly don't think they are very important.

A: Thanks, revised. We agree with the reviewer that strengthen our findings further. Following the reviewer’s suggestions, the following paragraph has been revised at the Results section as follows (pp. 20-21):

“In the sublinear PA regime, new nodes (i.e., inventors) are connected to old ones with a probability proportional to a fractional power of their degree. In this case, the network becomes a gel-like where every node is connected to all other nodes and degree distribution is the stretched exponential rather than power-law [6]. This asymptotic degree distribution for sublinearly growing networks is characterized by an exponent smaller than one and a maximum degree that scales as a power of the logarithm of the number of nodes [7]. Therefore, our results from using large datasets support Hypothesis H3, that is, the sublinear regime, and this is consistent with previous measurements on scientific collaboration [6, 8]. This also indicates that a perfect PA function (that is, the chance that an inventor gets a new collaborator is proportional to their current number of collaborators) is not supported in our study.”

The following paragraph has been changed at the Conclusion and Discussion section (pp. 26-27):

“Third, all the inventor networks considered here grew at a lower PA rate than in other collaboration networks such as scientific papers and patent citations [6]. We can infer that the observed distributions fit in a growth model in which the source of added edges are chosen according to a sublinear PA, but the destination is selected at random [9]. This is indicative of an inverse Matthew effect or a cumulative disadvantage and implies that past patent activity is not necessarily correlated with whatever growth mechanism is actually at play [10]. Therefore, we cannot assume that there is a clear growth mechanism for making new connections for inventors with a high degree [9]. It is also possible that the innovation system is not yet at full capacity in terms of the number of players necessary to make it sustainable or self-organizing [11]. Self-organization may be among the most important mechanisms leading to scale invariance and decreased system entropy, which are the foundations of evolution in complex systems [10]. This sublinear PA regime seems to resonate deeply with the expected restrictions on the behavior of inventors in a business setting, in which inventors are required to first seek the collaboration of (or are coerced to collaborate with) others within their firms and corporate alliances before accessing new knowledge outside their organizational boundaries [12]. Patent collaboration is a complex task and company policies seem to play a decisive role, especially in shaping the partner selection of inventor teams and therefore collaboration networks. This is increasingly the case as the increasing regulatory patent protection system tends to restrict rather than advance knowledge spillover and sharing [13].”

And also (pp. 28-29):

“The smallest technology groups, namely invention technologies in the fields of D. Textile and paper and E. Construction do not fit the gel point rules. Although there were indications of an early phase aggregation process, these inventor networks were still highly fragmented and much smaller in size than any of the other technology networks studied here, and this had important implications for their network topology. The creation of a complex structure as a result of the dynamics of the system is still pending in these sectors, as it is evident that the organization is still notably relegated in comparison with that of the other’s sectors.”

We hope that with the aforementioned changes we have addressed all issues mentioned in the reviewers’ comments. 

Sincerely, 

The authors.

 

References

1. Shannon P, Markiel A, Ozier O, Baliga NS, Wang JT, Ramage D, et al. Cytoscape: a software environment for integrated models of biomolecular interaction networks. Genome Res. 2003;13(11):2498-504. doi: doi: 10.1101/gr.1239303.

2. Kohl M, Wiese S, Warscheid B. Cytoscape: Software for Visualization and Analysis of Biological Networks. In: Hamacher M, Eisenacher M, Stephan C, editors. Data Mining in Proteomics: From Standards to Applications. Totowa, NJ: Humana Press; 2011. p. 291-303.

3. Ribeiro LC, Rapini MS, Silva LA, Albuquerque EM. Growth patterns of the network of international collaboration in science. Scientometrics. 2018;114(1):159-79. doi: 10.1007/s11192-017-2573-x.

4. McGlohon M, Akoglu L, Faloutsos C. Statistical Properties of Social Networks. In: Aggarwal CC, editor. Social network data analytics: Springer; 2011. p. 17-42.

5. Newman ME. The Structure and Function of Complex Networks. SIAM Review. 2003;45(2):167 - 256. PubMed PMID: doi:10.1137/S003614450342480.

6. Golosovsky M. Mechanisms of Complex Network Growth: Synthesis of the Preferential Attachment and Fitness Models. Physical Review E. 2018;97(6):062310. doi: 10.1103/physreve.97.062310.

7. Jog V, Loh P. Analysis of Centrality in Sublinear Preferential Attachment Trees via the Crump-Mode-Jagers Branching Process. IEEE Transactions on Network Science and Engineering. 2017;4(1):1-12. doi: 10.1109/TNSE.2016.2622923.

8. Wagner CS, Leydesdorff L. Network structure, self-organization, and the growth of international collaboration in science. Research Policy. 2005;34(10):1608-18.

9. Newman ME, Forrest S, Balthrop J. Email networks and the spread of computer viruses. Physical Review E. 2002;66(3):035101.

10. Katz JS. What Is a Complex Innovation System? PLOS ONE. 2016;11(6):e0156150. doi: 10.1371/journal.pone.0156150.

11. Perc M. Growth and structure of Slovenia’s scientific collaboration network. Journal of Informetrics. 2010;4(4):475-82. doi: https://doi.org/10.1016/j.joi.2010.04.003.

12. Crescenzi R, Nathan M, Rodríguez-Pose A. Do inventors talk to strangers? On proximity and collaborative knowledge creation. Research Policy. 2016;45(1):177-94.

13. Ma Y, Chi Q, Song L. Revealing structural patterns of patent citation by a two-boundary network model based on USPTO data. IEEE Access. 2020;8:23324-35. doi: 10.1109/ACCESS.2020.2969654.

---

## [Editor Report · Decision Letter 2]

20 Aug 2021

Exploring the topology and dynamic growth properties of co-invention networks and technology fields

PONE-D-20-41087R2

Dear Dr. PINTO,

We’re pleased to inform you that your manuscript has been judged scientifically suitable for publication and will be formally accepted for publication once it meets all outstanding technical requirements.

Kind regards,

Petter Holme, Ph.D.

Academic Editor

PLOS ONE
---

## [Editor Report · Acceptance letter]

24 Aug 2021

PONE-D-20-41087R2 

Exploring the topology and dynamic growth properties of co-invention networks and technology fields 

Dear Dr. Pinto:

I'm pleased to inform you that your manuscript has been deemed suitable for publication in PLOS ONE. Congratulations! Your manuscript is now with our production department. 

Kind regards, 

on behalf of

Dr. Petter Holme 

Academic Editor

PLOS ONE